# Robust Sub-Gaussian Principal Component Analysis and Width-Independent Schatten Packing

**Arun Jambulapati**
Stanford University
jmblpati@stanford.edu

**Jerry Li**
Microsoft Research
jerrl@microsoft.com

**Kevin Tian**
Stanford University
kjtian@stanford.edu

## Abstract

We develop two methods for the following fundamental statistical task: given an $\epsilon$-corrupted set of $n$ samples from a $d$-dimensional sub-Gaussian distribution, return an approximate top eigenvector of the covariance matrix. Our first robust PCA algorithm runs in polynomial time, returns a $1 - O(\epsilon \log \epsilon^{-1})$-approximate top eigenvector, and is based on a simple iterative filtering approach. Our second, which attains a slightly worse approximation factor, runs in nearly-linear time and sample complexity under a mild spectral gap assumption. These are the first polynomial-time algorithms yielding non-trivial information about the covariance of a corrupted sub-Gaussian distribution without requiring additional algebraic structure of moments. As a key technical tool, we develop the first width-independent solvers for Schatten-$p$ norm packing semidefinite programs, giving a $(1 + \epsilon)$-approximate solution in $O(p \log(\frac{nd}{\epsilon})\epsilon^{-1})$ input-sparsity time iterations (where $n$, $d$ are problem dimensions).

## 1 Introduction

We study two natural, but seemingly unrelated, problems in high dimensional robust statistics and continuous optimization respectively. As we will see, these problems have an intimate connection.

**Problem 1: Robust sub-Gaussian principal component analysis.** We consider the following statistical task, which we call *robust sub-Gaussian principal component analysis* (PCA). Given samples $X_1, \ldots, X_n$ from sub-Gaussian[1] distribution $\mathcal{D}$ with covariance $\boldsymbol{\Sigma}$, an $\epsilon$ fraction of which are arbitrarily corrupted, the task asks to output unit vector $u$ with $u^\top \boldsymbol{\Sigma} u \geq (1 - \gamma) \|\boldsymbol{\Sigma}\|_\infty$[2] for tolerance $\gamma$. Ergo, the goal is to robustly return a $(1 - \gamma)$-approximate top eigenvector of the covariance of sub-Gaussian $\mathcal{D}$. This is the natural extension of PCA to the robust statistics setting.

There has been a flurry of recent work on efficient algorithms for robust statistical tasks, e.g. covariance estimation and PCA. From an information-theoretic perspective, sub-Gaussian concentration suffices for robust covariance estimation. Nonetheless, to date all polynomial-time algorithms achieving nontrivial guarantees on covariance estimation (including PCA specifically) in the presence of adversarial noise require additional algebraic structure. For instance, sum-of-squares certifiably bounded moments have been leveraged in polynomial time covariance estimation algorithms [HL18, KSS18]; however, this is a stronger assumption than sub-Gaussianity.

In many applications (see discussion in [DKK+17]), the end goal of covariance estimation is PCA. Thus, a natural question which relaxes robust covariance estimation is: can we robustly estimate the top eigenvector of the covariance $\boldsymbol{\Sigma}$, assuming only sub-Gaussian concentration? Our work answers this question affirmatively via two incomparable algorithms. The first achieves $\gamma = O(\epsilon \log \epsilon^{-1})$ in

polynomial time; the second achieves $\gamma = O(\sqrt{\epsilon \log \epsilon^{-1} \log d})$, in nearly-linear time under a mild gap assumption on $\Sigma$. Moreover, both methods have nearly-optimal sample complexity.

**Problem 2: Width-independent Schatten packing.** We consider a natural generalization of packing semidefinite programs (SDPs) which we call *Schatten packing*. Given symmetric positive semidefinite $\mathbf{A}_1, \ldots, \mathbf{A}_n$ and parameter $p \geq 1$, a Schatten packing SDP asks to solve the optimization problem

$$\min \left\| \sum_{i \in [n]} w_i \mathbf{A}_i \right\|_p \quad \text{subject to } w \in \Delta^n. \tag{1}$$

Here, $\|\mathbf{M}\|_p$ is the Schatten-$p$ norm of matrix $\mathbf{M}$ and $\Delta^n$ is the probability simplex (see Section 2). When $p = \infty$, (1) is the well-studied (standard) packing SDP objective [JY11, ALO16, PTZ16], which asks to find the most spectrally bounded convex combination of packing matrices. For smaller $p$, the objective encourages combinations more (spectrally) uniformly distributed over directions.

The specialization of (1) to diagonal matrices is a smooth generalization of packing linear programs, previously studied in the context of fair resource allocation [MSZ16, DFO18]. For the $\ell_\infty$ case of (1), packing SDPs have the desirable property of admitting "width-independent" approximation algorithms via exploiting positivity structure. Specifically, width-independent solvers obtain multiplicative approximations with runtimes independent or logarithmically dependent on size parameters of the problem. This is a strengthening of additive notions of approximation typically used for approximate semidefinite programming. Our work gives the first width-independent solver for Schatten packing.

## 1.1 Previous work

**Learning with adversarial outliers.** The study of estimators robust to a small fraction of adversarial outliers dates back to foundational work, e.g. [Hub64, Tuk75]. Following more recent work [LRV16, DKK+19], there has been significant interest in efficient, robust algorithms for statistical tasks in high-dimensional settings. We focus on methods robustly estimating covariance properties here, and defer a thorough discussion of the (extensive) robust statistics literature to [Ste18, Li18, DK19].

There has been quite a bit of work in understanding and giving guarantees for robust covariance estimation where the uncorrupted distribution is exactly Gaussian [DKK+17, DKK+18, DKK+19, CDGW19]. These algorithms strongly use relationships between higher-order moments of Gaussian distributions via Isserlis' theorem. Departing from the Gaussian setting, work of [LRV16] showed that if the distribution is an affine transformation of a 4-wise independent distribution, robust covariance estimation is possible. This was extended by [KSS18], which also assumed nontrivial structure in the moments of the distribution, namely that sub-Gaussianity was certifiable via the sum-of-squares proof system. To the best of our knowledge it has remained open to give nontrivial guarantees for robust estimation of any covariance properties under minimal assumptions, i.e. sub-Gaussian concentration.

All aforementioned algorithms also yield guarantees for robust PCA, by applying a top eigenvector method to the learned covariance. However, performing robust PCA via the intermediate covariance estimation step is lossy, both statistically and computationally. From a statistical perspective, $\Omega(d^2)$ samples are necessary to learn the covariance of a $d$-dimensional Gaussian in Frobenius norm (and for known efficient algorithms for spectral norm error [DKS17]); in contrast, $O(d)$ samples suffice for (non-robust) PCA. Computationally, even when the underlying distrubution is exactly Gaussian, the best-known covariance estimation algorithms run in time $\tilde{\Omega}(d^{3.25})$; algorithms working in more general settings based on the sum-of-squares approach require much more time. In contrast, the power method for PCA in a $d \times d$ matrix takes time $\tilde{O}(d^2)^3$. Motivated by this, our work initiates the direct study of robust PCA, which is often independently interesting in applications.

We remark there is another problem termed "robust PCA" in the literature, e.g. [CLMW11], under a different generative model. We defer a detailed discussion to [DKK+17], which experimentally shows that algorithms from that line of work do not transfer well to our corruption model.

**Width-independent iterative methods.** Semidefinite programming (SDP) and its linear programming specialization are fundamental computational tasks, with myriad applications in learning, operations research, and computer science. Though general-purpose polynomial time algorithms exist for

SDPs ([NN94]), in practical settings in high dimensions, approximations depending linearly on input size and polynomially on error $\epsilon$ are sometimes desirable. To this end, approximation algorithms based on entropic mirror descent have been intensely studied [WK06, AK16, GHM15, AL17, CDST19], obtaining $\epsilon$ additive approximations to the objective with runtimes depending polynomially on $\rho/\epsilon$, where $\rho$ is the "width", the largest spectral norm of a constraint.

For structured SDPs, stronger guarantees can be obtained in terms of width. Specifically, several algorithms developed for packing SDPs ((1) with $p = \infty$) yield $(1+\epsilon)$-*multiplicative* approximations to the objective, with *logarithmic* dependence on width [JY11, PTZ16, ALO16, JLL$^+$20]. As $\rho$ upper bounds objective value in this setting, in the worst case runtimes of width-dependent solvers yielding $\epsilon\rho$-additive approximations have similar dependences as width-independent counterparts. Width-independent solvers simultaneously yield stronger multiplicative bounds at all scales of objective value, making them desirable in suitable applications. In particular, $\ell_\infty$ packing SDPs have found great utility in robust statistics algorithm design [CG18, CDG19, CDGW19, DL19].

Beyond $\ell_\infty$ packing, width-independent guarantees in the SDP literature are few and far between; to our knowledge, other than the covering and mixed solvers of [JLL$^+$20], ours is the first such guarantee for a broader family of objectives[4]. Our method complements analogous $\ell_p$ extensions in the width-dependent setting, e.g. [ALO15], as well as width-independent solvers for $\ell_p$ packing linear programs [MSZ16, DFO18]. We highlight the fair packing solvers of [MSZ16, DFO18], motivated by problems in equitable resource allocation, which further solved $\ell_p$ packing variants for $p \notin [1, \infty)$. We find analogous problems in semidefinite settings interesting, and defer to future work.

**Concurrent work.** Concurrent work by Kong et al. [KSKO20] also develops a PCA algorithm tolerant to a bounded fraction of adversarial corruption. Their method is similar to our algorithm based on soft downweighting (Algorithm 6), is analyzed under a fourth moment bound assumption (as opposed to sub-Gaussianity as in this paper), and also generalizes to top-$k$ eigenvector estimation. To our knowledge, our fast algorithm (Algorithm 4) is the first in the literature which robustly solves the 1-PCA problem in near-linear time (for gapped covariances), at the cost of weaker error bounds.

## 1.2 Our results

**Robust sub-Gaussian principal component analysis.** We give two algorithms for robust sub-Gaussian PCA[5]. Both are sample optimal, polynomial-time, and assume only sub-Gaussianity. The first is via a simple filtering approach, as summarized in the following (and developed in Section 3).

**Theorem 1.** *Under Assumption 1, let $\delta \in [0, 1]$, and $n = \Omega\left(\frac{d + \log \delta^{-1}}{(\epsilon \log \epsilon^{-1})^2}\right)$. Algorithm 6 runs in time $O(\frac{nd^2}{\epsilon} \log \frac{n}{\delta\epsilon} \log \frac{n}{\delta})$, and outputs $u$ with $u^\top \Sigma u > (1 - C^\star \epsilon \log \epsilon^{-1}) \|\Sigma\|_\infty$, for $C^\star$ a fixed multiple of parameter $c$ in Assumption 1, with probability at least $1 - \delta$.*

Our second algorithm is more efficient under mild conditions, but yields a worse approximation $1 - \gamma$ for $\gamma = O(\sqrt{\epsilon \log \epsilon^{-1} \log d})$. Specifically, if there are few eigenvalues of $\Sigma$ larger than $1 - \gamma$, our algorithm runs in nearly-linear time. Note that if there are many eigenvalues above this threshold, then the PCA problem itself is not very well-posed; our algorithm is very efficient in the interesting setting where the approximate top eigenvector is identifiable. We state our main algorithmic guarantee here, and defer details to Section 5.

**Theorem 2.** *Under Assumption 1, let $\delta \in [0, 1]$, $n = \Omega\left(\frac{d + \log \delta^{-1}}{(\epsilon \log \epsilon^{-1})^2}\right)$, $\gamma = C\sqrt{\epsilon \log \epsilon^{-1} \log d}$, for $C$ a fixed multiple of parameter $c$ from Assumption 1, and let $t \in [d]$ satisfy $\Sigma_{t+1} < (1 - \gamma) \|\Sigma\|_\infty$. Algorithm 4 outputs a unit vector $u \in \mathbb{R}^d$ with $u^\top \Sigma u \geq (1 - \gamma) \|\Sigma\|_\infty$ in time $\widetilde{O}(\frac{nd}{\epsilon^{4.5}} + \frac{ndt}{\epsilon^{1.5}})$.*

Since $\Omega(d\epsilon^{-2})$ samples are necessary for a $(1 - \epsilon)$-approximation to the top eigenvector of $\Sigma$ via uncorrupted samples, our first method is sample-optimal, as is our second up to a $\widetilde{O}(\epsilon^{-1})$ factor.

**Width-independent Schatten packing.** Our second method crucially requires an efficient solver for Schatten packing SDPs. We demonstrate that Schatten packing, i.e. (1) for arbitrary $p$, admits width-independent solvers. We state an informal guarantee, and defer details to Section 4.

**Theorem 3.** *Let $\{\mathbf{A}_i\}_{i \in [n]} \in \mathbb{S}_{\geq 0}^d$, and $\epsilon > 0$. There is an algorithm taking $O(\frac{p \log(\frac{nd}{\epsilon})}{\epsilon})$ iterations, returning a $1 + \epsilon$ multiplicative approximation to the problem (1). For odd $p$, each iteration can be implemented in time nearly-linear in the number of nonzeros amongst all $\{\mathbf{A}_i\}_{i \in [n]}$.*

## 2 Preliminaries

**General notation.** $[n]$ denotes the set $1 \leq i \leq n$. The operation $\circ$ applied to two vectors of equal dimension is their entrywise product. Applied to a vector, $\|\cdot\|_p$ is the $\ell_p$ norm; applied to a symmetric matrix, $\|\cdot\|_p$ is the Schatten-$p$ norm, i.e. the $\ell_p$ norm of the spectrum. The *dual norm* of $\ell_p$ is $\ell_q$ for $q = \frac{p}{p-1}$; when $p = \infty$, $q = 1$. $\Delta^n$ is the $n$-dimensional simplex (subset of positive orthant with $\ell_1$-norm 1) and we define $\mathfrak{S}_\varepsilon^n \subseteq \Delta^n$ to be the truncated simplex:

$$\mathfrak{S}_\varepsilon^n := \left\{ w \in \mathbb{R}_{\geq 0}^n \;\middle|\; \|w\|_1 = 1, \; w \leq \frac{1}{n(1-\varepsilon)} \text{ entrywise} \right\}. \tag{2}$$

**Matrices.** $\mathbb{S}^d$ is $d \times d$ symmetric matrices, and $\mathbb{S}_{\geq 0}^d$ is the positive semidefinite subset. $\mathbf{I}$ is the identity of appropriate dimension. $\lambda_{\max}$, $\lambda_{\min}$, and $\mathrm{Tr}$ are the largest and smallest eigenvalues and trace of a symmetric matrix. For $\mathbf{M}, \mathbf{N} \in \mathbb{S}^d$, $\langle \mathbf{M}, \mathbf{N} \rangle := \mathrm{Tr}\,(\mathbf{MN})$ and we use the Loewner order $\preceq$, ($\mathbf{M} \preceq \mathbf{N}$ iff $\mathbf{N} - \mathbf{M} \in \mathbb{S}_{\geq 0}^d$). The seminorm of $\mathbf{M} \succeq 0$ is $\|v\|_{\mathbf{M}} := \sqrt{v^\top \mathbf{M} v}$.

**Fact 1.** *For $\mathbf{A}$, $\mathbf{B}$ with compatible dimension, $\mathrm{Tr}(\mathbf{AB}) = \mathrm{Tr}(\mathbf{BA})$. For $\mathbf{M}, \mathbf{N} \in \mathbb{S}_{\geq 0}^d$, $\langle \mathbf{M}, \mathbf{N} \rangle \geq 0$.*

**Fact 2.** *We have the following characterization of the Schatten-$p$ norm: for $\mathbf{M} \in \mathbb{S}^d$, and $q = \frac{p}{p-1}$,*

$$\|\mathbf{M}\|_p = \sup_{\mathbf{N} \in \mathbb{S}^d, \, \|\mathbf{N}\|_q = 1} \langle \mathbf{N}, \mathbf{M} \rangle.$$

*For $\mathbf{M} = \sum_{j \in [d]} \lambda_i v_i v_i^\top$, the satisfying $\mathbf{N}$ is $\frac{\sum_{j \in [d]} \pm \lambda_i^{p-1} v_i v_i^\top}{\|\mathbf{M}\|_p^{p-1}}$, so $\mathbf{NM}$ has spectrum $\frac{|\lambda|^p}{\|\mathbf{M}\|_p^{p-1}}$.*

**Distributions.** We denote drawing vector $X$ from distribution $\mathcal{D}$ by $X \sim \mathcal{D}$, and the covariance $\boldsymbol{\Sigma}$ of $\mathcal{D}$ is $\mathbb{E}_{X \sim \mathcal{D}}\left[ X X^\top \right]$. We say scalar distribution $\mathcal{D}$ is $\gamma^2$-sub-Gaussian if $\mathbb{E}_{X \sim \mathcal{D}}[X] = 0$ and

$$\mathbb{E}_{X \sim \mathcal{D}}\left[ \exp\left(tX\right) \right] \leq \exp\left( \frac{t^2 \gamma^2}{2} \right) \; \forall t \in \mathbb{R}.$$

Multivariate $\mathcal{D}$ has sub-Gaussian proxy $\boldsymbol{\Gamma}$ if its restriction to any unit $v$ is $\|v\|_{\boldsymbol{\Gamma}}^2$-sub-Gaussian, i.e.

$$\mathbb{E}_{X \sim \mathcal{D}}\left[ \exp\left(tX^\top v\right) \right] \leq \exp\left( \frac{t^2 \|v\|_{\boldsymbol{\Gamma}}^2}{2} \right) \text{ for all } \|v\|_2 = 1, \; t \in \mathbb{R}. \tag{3}$$

We consider the following standard model for gross corruption with respect to a distribution $\mathcal{D}$.

**Assumption 1** (Corruption model, see [DKK+19]). *Let $\mathcal{D}$ be a mean-zero distribution on $\mathbb{R}^d$ with covariance $\boldsymbol{\Sigma}$ and sub-Gaussian proxy $\boldsymbol{\Gamma} \preceq c\boldsymbol{\Sigma}$ for a constant $c$. Denote by index set $G'$ with $|G'| = n$ a set of (uncorrupted) samples $\{X_i\}_{i \in G'} \sim \mathcal{D}$. An adversary arbitrarily replaces $\epsilon n$ points in $G'$; we denote the new index set by $[n] = B \cup G$, where $B$ is the (unknown) set of points added by an adversary, and $G \subseteq G'$ is the set of points from $G'$ that were not changed.*

As we only estimate covariance properties, the assumption that $\mathcal{D}$ is mean-zero only loses constants in problem parameters, by pairing samples and subtracting them (cf. [DKK+19], Section 4.5.1).

## 3 Robust sub-Gaussian PCA via filtering

In this section, we sketch the proof of Theorem 1, which gives guarantees on our filtering algorithm for robust sub-Gaussian PCA. This algorithm obtains stronger statistical guarantees than Theorem 2, at the cost of super-linear runtime; the algorithm is given as Algorithm 6. Our analysis stems largely from concentration facts about sub-Gaussian distributions, as well as the following (folklore) fact regarding estimation of variance along any particular direction.

**Lemma 1.** *Under Assumption 1, let $\delta \in [0,1]$, $n = \Omega\left(\frac{\log \delta^{-1}}{(\epsilon \log \epsilon^{-1})^2}\right)$, and $u \in \mathbb{R}^d$ be a fixed unit vector. Algorithm 5,* 1DRobustVariance, *takes input $\{X_i\}_{i \in [n]}$, $u$, and $\epsilon$, and outputs $\sigma_u^2$ with $|u^\top \Sigma u - \sigma_u^2| < C u^\top \Sigma u \cdot \epsilon \log \epsilon^{-1}$ with probability at least $1 - \delta$, and runs in time $O(nd + n \log n)$, for $C$ a fixed multiple of the parameter $c$ in Assumption 1.*

In other words, we show that using corrupted samples, we can efficiently estimate a $1 + O(\epsilon \log \epsilon^{-1})$-multiplicative approximation of the variance of $\mathcal{D}$ in any unit direction[6]. This proof is deferred to Appendix B for completeness. Algorithm 6 combines this key insight with a soft filtering approach which has found many applications in the recent robust statistics literature, suggested by the following known structural fact found in previous work (e.g. Lemma A.1 of [DHL19], see also [SCV17, Ste18]).

**Lemma 2.** *Let $\{a_i\}_{i \in [m]}$, $\{w_i\}_{i \in [m]}$ be sets of nonnegative reals, and $a_{\max} = \max_{i \in [m]} a_i$. Define $w_i' = \left(1 - \frac{a_i}{a_{\max}}\right) w_i$, for all $i \in [m]$. Consider any disjoint partition $I_B$, $I_G$ of $[m]$ with $\sum_{i \in I_B} w_i a_i > \sum_{i \in I_G} w_i a_i$. Then, $\sum_{i \in I_B} w_i - w_i' > \frac{1}{2a_{\max}} \sum_{i \in [m]} w_i a_i > \sum_{i \in I_G} w_i - w_i'$.*

Our Algorithm 6, PCAFilter, takes as input a set of corrupted samples $\{X_i\}_{i \in [n]}$ following Assumption 1 and the corruption parameter $\epsilon$. At a high level, it initializes a uniform weight vector $w^{(0)}$, and iteratively operates as follows (we denote by $\mathbf{M}(w)$ the empirical covariance $\sum_{i \in [n]} w_i X_i X_i^\top$).

1. $u_t \leftarrow$ approximate top eigenvector of $\mathbf{M}(w^{(t-1)})$ via power iteration.
2. Compute $\sigma_t^2 \leftarrow$ 1DRobustVariance($\{X_i\}_{i \in [n]}, u_t, \epsilon$).
3. If $\sigma_t^2 > (1 - O(\epsilon \log \epsilon^{-1})) \cdot u_t^\top \mathbf{M}(w^{(t-1)}) u_t$, then terminate and return $u_t$.
4. Else:
   (a) Sort indices $i \in [n]$ by $a_i \leftarrow \langle u_t, X_i \rangle^2$, with $a_1$ smallest.
   (b) Let $\ell \leq i \leq n$ be the smallest set for which $\sum_{i=\ell}^n w_i \geq 2\epsilon$, and apply the downweighting procedure of Lemma 2 to this subset of indices.

The analysis of Algorithm 6 then proceeds in two stages.

**Monotonicity of downweighting.** We show the invariant criteria for Lemma 2 (namely, that for the set $\ell \leq i \leq n$ in every iteration, there is more spectral mass on bad points than good) holds inductively for our algorithm. Specifically, lack of termination implies $\mathbf{M}(w^{(t-1)})$ puts significant mass on bad directions, which combined with concentration of good directions yields the invariant. The details of this argument can be found as Lemma 11.

**Roughly uniform weightings imply approximation quality.** As Lemma 2 then applies, the procedure always removes more mass from bad points than good, and thus can only remove at most $2\epsilon$ mass total by the corruption model. Thus, the weights $w^{(t)}$ are always roughly uniform (in $\mathfrak{S}_{O(\epsilon)}^n$), which by standard concentration facts (see Appendix A) imply the quality of the approximate top eigenvector is good. Moreover, the iteration count is bounded by roughly $d$ because whenever the algorithm does not terminate, enough mass is removed from large spectral directions. Combining with the termination criteria imply that when a vector is returned, it is a close approximation to the top direction of $\Sigma$. Details can be found as Lemma 13 and in the proof of Theorem 1.

# 4 Schatten packing

For our second robust PCA algorithm, developed in Section 5, we require a key technical tool which we now develop in this section. The tool, Schatten-norm packing semidefinite programs (and hybrid-norm extensions), is a smoothed generalization of the classical packing semidefinite program, which may be of independent interest in other applications. At a high level, the reason Schatten packing solvers are useful for the robust PCA problem is because while an adversary can fool a PCA algorithm based on operator-norm semidefinite programs by "promoting" a single other eigenvector to have a larger variance, a $p$-norm-based semidefinite program forces a tradeoff between the number of directions promoted and the amount of variance introduced.

## 4.1 Mirror descent interpretation of [MRWZ16]

We begin by reinterpreting the [MRWZ16] solver, which achieves the state-of-the-art parallel runtime for packing LPs[7]. An ($\ell_\infty$) packing LP algorithm solves the following decision problem.[8]

**Problem 1** ($\ell_\infty$ packing linear program). *Given entrywise nonnegative $\mathbf{A} \in \mathbb{R}_{\geq 0}^{d \times n}$, either find primal solution $x \in \Delta^n$ with $\|\mathbf{A}x\|_\infty \leq 1 + \epsilon$ or dual solution $y \in \Delta^d$ with $\mathbf{A}^\top y \geq (1 - \epsilon)\mathbf{1}$.*

---

**Algorithm 1** PackingLP($\mathbf{A}, \epsilon$)

1: **Input:** $\mathbf{A} \in \mathbb{R}_{\geq 0}^{d \times n}$, $\epsilon \in [0, \frac{1}{2}]$
2: $K \leftarrow \frac{3\log(d)}{\epsilon}$, $\eta \leftarrow K^{-1}$, $T \leftarrow \frac{4\log(d)\log(nd/\epsilon)}{\epsilon^2}$
3: $[w_0]_i \leftarrow \frac{\epsilon}{n^2 d}$ for all $i \in [n]$, $z \leftarrow \mathbf{0}$, $t \leftarrow 0$
4: **while** $\mathbf{A}w_t \leq K\mathbf{1}$, $\|w_t\|_1 \leq K$ **do**
5: $\quad v_t \leftarrow \frac{\exp(\mathbf{A}w_t)}{\|\exp(\mathbf{A}w_t)\|_1}$
6: $\quad g_t \leftarrow \max(0, \mathbf{1} - \mathbf{A}^\top v_t)$ entrywise
7: $\quad w_{t+1} \leftarrow w_t \circ (1 + \eta g_t)$, $z \leftarrow z + v_t$, $t \leftarrow t + 1$
8: $\quad$ **if** $t \geq T$ **then**
9: $\quad\quad$ **return** $y \leftarrow \frac{1}{T}z$
10: $\quad$ **end if**
11: **end while**
12: **return** $x \leftarrow \frac{w_t}{\|w_t\|_1}$

---

The following result is shown in [MRWZ16].

**Proposition 1.** PackingLP *(Algorithm 1) solves Problem 1 in* $O(\text{nnz}(\mathbf{A}) \cdot \frac{\log(d)\log(nd/\epsilon)}{\epsilon^2})$ *time.*

Our interpretation of the analysis of [MRWZ16] combines two ingredients: a potential argument and mirror descent (alternatively known as the "multiplicative weights" framework), which yields a dual feasible point if $\|w_t\|_1$ did not grow sufficiently.

**Potential argument.** The potential used by [MRWZ16] is $\log(\sum_{j \in [d]} \exp([\mathbf{A}w_t]_j)) - \|w_t\|_1$, well-known to be a $O(\log d)$-additive approximation of $\|\mathbf{A}w_t\|_\infty - \|w_t\|_1$. As soon as $\|\mathbf{A}w_t\|_\infty$ or $\|w_t\|_1$ reaches the scale $O(\frac{\log d}{\epsilon})$, by nonnegativity this becomes a multiplicative guarantee, motivating the setting of threshold $K$. To prove the potential is monotone, [MRWZ16] uses step size $K^{-1}$ and a Taylor approximation; combining with the termination condition yields the desired claim.

**Mirror descent.** To certify that $w_t$ grows sufficiently (e.g. the method terminates in few iterations, else dual feasibility holds), we interpret the step $w_{t+1} \leftarrow w_t \circ (1 + \eta g_t)$ as approximate entropic mirror descent. Specifically, we track the quantity $\sum_{0 \leq t < T} \langle \eta g_t, u \rangle$, and show that if $\|w_t\|_1$ has not grown sufficiently, then it must be bounded for every $u \in \Delta^n$, certifying dual feasibility. Formally, for any $g_t$ sequence and $u \in \Delta^n$, we show

$$O(\log(nd/\epsilon)) + \log\left(\frac{\|w_T\|_1}{\|w_0\|_1}\right) \geq \sum_{0 \leq t < T} \langle \eta g_t, u \rangle \geq \eta \sum_{0 \leq t < T} \langle \mathbf{1} - \mathbf{A}^\top v_t, u \rangle.$$

The last inequality followed by $g_t$ being an upwards truncation. If $\|w_T\|_1$ is bounded (else, we have primal feasibility), we show the entire above expression is bounded $O(\log \frac{nd}{\epsilon})$ for any $u$. Thus, by setting $T = O(\frac{\log(nd/\epsilon)}{\eta\epsilon})$ and choosing $u$ to be each coordinate indicator, it follows that the average of all $v_t$ is coordinatewise at least $1 - \epsilon$, and solves Problem 1 as a dual solution.

Our $g_t$ is the (truncated) gradient of the function used in the potential analysis, so its form allows us to interpret dual feasibility (e.g. $v_t$ has $\ell_1$ norm 1 and is a valid dual point). Our analysis patterns standard mirror descent, complemented by side information which says that lack of a primal solution can transform a regret guarantee into a feasibility bound. We apply this framework to analyze $\ell_p$

variants of Problem 1, via different potentials; our proofs are quite straightforward upon adopting this perspective, and we believe it may yield new insights for instances with positivity structure.

## 4.2  $\ell_p$-norm packing linear programs

In this section, we give an example of the framework proposed in Section 4.1, for approximately solving $\ell_p$ norm packing linear programs. Specifically, we now consider the generalization of Problem 1 to $\ell_p$ norms; throughout, $q = \frac{p}{p-1}$ is the dual norm.

**Problem 2** ($\ell_p$ packing linear program). *Given entrywise nonnegative $\mathbf{A} \in \mathbb{R}_{\geq 0}^{d \times n}$, either find primal solution $x \in \Delta^n$ with $\|\mathbf{A}x\|_p \leq 1 + \epsilon$ or dual solution $y \in \mathbb{R}_{\geq 0}^d$, $\|y\|_q = 1$ with $\mathbf{A}^\top y \geq (1 - \epsilon)\mathbf{1}$.*

For $p = \frac{\log d}{\epsilon}$, Problem 2 recovers Problem 1 up to constants as $\ell_p$ multiplicatively approximates $\ell_\infty$ by $1 + \epsilon$. We now state our method for solving Problem 2 as Algorithm 2.

---

**Algorithm 2** PNormPacking$(\mathbf{A}, \epsilon, p)$

---

1: **Input:** $\mathbf{A} \in \mathbb{R}_{\geq 0}^{d \times n}, \epsilon \in [0, \frac{1}{2}], p \geq 2$
2: $\eta \leftarrow p^{-1}, T \leftarrow \frac{4p \log(\frac{nd}{\epsilon})}{\epsilon}$
3: $[w_0]_i \leftarrow \frac{\epsilon}{n^2 d}$ for all $i \in [n]$, $z \leftarrow \mathbf{0}, t \leftarrow 0$
4: **while** $\|w_t\|_1 \leq \epsilon^{-1}$ **do**
5:      $g_t \leftarrow \max(0, \mathbf{1} - \mathbf{A}^\top (v_t)^{p-1})$ entrywise, for $v_t \leftarrow \frac{\mathbf{A}w_t}{\|\mathbf{A}w_t\|_p}$
6:      $w_{t+1} \leftarrow w_t \circ (1 + \eta g_t), z \leftarrow z + (v_t)^{p-1}, t \leftarrow t + 1$
7:      **if** $t \geq T$ **then**
8:          **return** $y = \frac{z}{\|z\|_q}$
9:      **end if**
10: **end while**
11: **return** $x = \frac{w_t}{\|w_t\|_1}$

---

Other than changing parameters, the only difference from Algorithm 1 is that $v$ is a point with unit $\ell_q$ norm induced by the gradient of our potential $\Phi_t$. We state our main potential fact, whose proof is based straightforwardly on Taylor expanding $\|\cdot\|_p$, and deferred to Appendix C for brevity.

**Lemma 3.** *In all iterations $t$ of Algorithm 2, defining $\Phi_t := \|\mathbf{A}w_t\|_p - \|w_t\|_1$, $\Phi_{t+1} \leq \Phi_t$.*

We now state our main result, which leverages the potential bound following the framework of Section 4.1. A proof can be found in Appendix C.

**Theorem 4.** *Algorithm 2 runs in time $O(\text{nnz}(\mathbf{A}) \cdot \frac{p \log(nd/\epsilon)}{\epsilon})$. Further, its output solves Problem 2.*

## 4.3  Schatten-norm packing semidefinite programs

We generalize Algorithm 2 to solve Schatten packing semidefinite programs, which we now define.

**Problem 3.** *Given $\{\mathbf{A}_i\}_{i \in [n]} \in \mathbb{S}_{\geq 0}^d$, either find primal solution $x \in \Delta^n$ with $\left\|\sum_{i \in [n]} x_i \mathbf{A}_i\right\|_p \leq 1 + \epsilon$ or dual solution $\mathbf{Y} \in \mathbb{S}_{\geq 0}^d$, $\|\mathbf{Y}\|_q = 1$ with $\langle \mathbf{A}_i, \mathbf{Y} \rangle \geq 1 - \epsilon$ for all $i \in [n]$.*

We assume that $p$ is an odd integer for simplicity (sufficient for our applications), and leave for interesting future work the cases when $p$ is even or noninteger. The potential used in the analysis and an overall guarantee are stated here, and deferred to Appendix C. The proofs are simple modifications of Lemma 3 and Theorem 4 using trace inequalities (similar to those in [JLL+20]) in place of scalar inequalities, as well as efficient approximation of quantities in Line 5 via the standard technique of Johnson-Lindestrauss projections.

**Lemma 4.** *In all iterations $t$ of Algorithm 3, defining $\Phi_t := \left\|\sum_{i \in [n]} [w_t]_i \mathbf{A}_i\right\|_p - \|w_t\|_1$, $\Phi_{t+1} \leq \Phi_t$.*

---
**Algorithm 3** SchattenPacking($\{\mathbf{A}_i\}_{i\in[n]}, \epsilon, p$)
---
1: **Input:** $\{\mathbf{A}_i\}_{i\in[n]} \in \mathbb{S}_{\geq 0}^d, \epsilon \in [0, \frac{1}{2}], p \geq 2$
2: $\eta \leftarrow p^{-1}, T \leftarrow \frac{4p\log(\frac{nd}{\epsilon})}{\epsilon}$
3: $[w_0]_i \leftarrow \frac{\epsilon}{n^2 d}$ for all $i \in [n], z \leftarrow 0$
4: **while** $\|w_t\|_1 \leq \epsilon^{-1}$ **do**
5:     $g_t \leftarrow \max\left(0, \mathbf{1} - \left\langle \mathbf{A}_i, \mathbf{V}_t^{p-1}\right\rangle\right)$ entrywise, for $\mathbf{V}_t \leftarrow \frac{\sum_{i\in[n]}[w_t]_i \mathbf{A}_i}{\left\|\sum_{i\in[n]}[w_t]_i \mathbf{A}_i\right\|_p}$
6:     $w_{t+1} \leftarrow w_t \circ (1 + \eta g_t), \mathbf{Z} \leftarrow \mathbf{Z} + (\mathbf{V}_t)^{p-1}, t \leftarrow t+1$
7:     **if** $t \geq T$ **then**
8:        **return** $\mathbf{Y} = \frac{\mathbf{Z}}{\|\mathbf{Z}\|_q}$
9:     **end if**
10: **end while**
11: **return** $x = \frac{w_t}{\|w_t\|_1}$
---

**Theorem 5.** *Let $p$ be odd. Algorithm 3 runs in $O(\frac{p\log(nd/\epsilon)}{\epsilon})$ iterations, and its output solves Problem 3. Each iteration is implementable in $O(\mathrm{nnz} \cdot \frac{p\log(nd/\epsilon)}{\epsilon^2})$, where $\mathrm{nnz}$ is the number of nonzero entries amongst all $\{\mathbf{A}_i\}_{i\in[n]}$, losing $O(\epsilon)$ in the quality of Problem 3 with probability $1 - \mathrm{poly}((nd/\epsilon)^{-1})$.*

### 4.4 Schatten packing with a $\ell_\infty$ constraint

We remark that the framework outlined in Section 4.1 is flexible enough to handle mixed-norm packing problems. Specifically, developments in Section 5 require the following guarantee.

**Proposition 2.** *Following Theorem 5's notation, let $p$ be odd, $\{\mathbf{A}_i\}_{i\in[n]} \in \mathbb{S}_{\geq 0}^d, 0 < \epsilon = O(\alpha)$, and*

$$\min_{\substack{x\in\Delta^n \\ \|x\|_\infty \leq \frac{1+\alpha}{n}}} \|\mathcal{A}(x)\|_p = \mathrm{OPT}. \tag{4}$$

*for $\mathcal{A}(x) := \sum_{i\in[n]} x_i \mathbf{A}_i$. Given estimate of $\mathrm{OPT}$ exponentially bounded in $\frac{nd}{\epsilon}$, there is a procedure calling Algorithm 7 $O(\log \frac{nd}{\epsilon})$ times giving $x \in \Delta^n$ with $\|x\|_\infty \leq \frac{(1+\alpha)(1+\epsilon)}{n}$, $\|\mathcal{A}(x)\|_p \leq (1+\epsilon)\mathrm{OPT}$. Algorithm 7 runs in $O(\frac{\log(nd/\epsilon)\log n}{\epsilon^2})$ iterations, each requiring time $O(\mathrm{nnz} \cdot \frac{p\log(nd/\epsilon)}{\epsilon^2})$.*

Our method, found in Appendix C, approximately solves (4) by first applying a standard binary search to place $\mathcal{A}(x)$ on the right scale, for which it suffices to solve an approximate decision problem. Then, we apply a truncated mirror descent procedure on the potential $\Phi(w) = \log(\exp(\|\mathcal{A}(w)\|_p) + \exp(\frac{n}{1+\alpha}\|w\|_\infty)) - \|w\|_1$, and prove correctness for solving the decision problem following the framework we outlined in Section 4.1.

## 5 Robust sub-Gaussian PCA in nearly-linear time

We give our nearly-linear time robust PCA method, leveraging developments of Section 4. Throughout, we will be operating under Assumption 1, for some corruption parameter $\epsilon$ with $\epsilon \log \epsilon^{-1} \log d = O(1)$; $\epsilon = O(\frac{1}{\log d \log\log d})$ suffices. We now develop tools to prove Theorem 2.

Algorithm 4 uses three subroutines: our earlier 1DRobustVariance method (Lemma 1), an application of our earlier Proposition 2 to approximate the solution to

$$\min_{w\in\mathfrak{S}_\epsilon^n} \left\|\sum_{i\in[n]} w_i X_i X_i^\top\right\|_p, \text{ for } p = \Theta\left(\sqrt{\frac{\log d}{\epsilon\log\epsilon^{-1}}}\right), \tag{5}$$

and a method for computing approximate eigenvectors by [MM15] (discussed in Appendix D).

**Proposition 3.** *There is an algorithm* Power *(Algorithm 1, [MM15]), parameterized by $t \in [d]$, tolerance $\tilde{\epsilon} > 0$, $p \geq 1$, and $\mathbf{A} \in \mathbb{S}_{\geq 0}^d$, which outputs orthonormal $\{z_j\}_{j \in [t]}$ with the guarantee*

$$\left.\begin{array}{ll} \left| z_j^\top \mathbf{A}^p z_j - \lambda_j^p(\mathbf{A}) \right| & \leq \tilde{\epsilon} \lambda_j^p(\mathbf{A}) \\ \left| z_j^\top \mathbf{A}^{p-1} z_j - \lambda_j^{p-1}(\mathbf{A}) \right| & \leq \tilde{\epsilon} \lambda_j^{p-1}(\mathbf{A}) \end{array}\right\} \text{ for all } j \in [t]. \tag{6}$$

*Here, $\lambda_j(\mathbf{A})$ is the $j^{th}$ largest eigenvalue of $\mathbf{A}$. The total time required by the method is $O(\text{nnz}(\mathbf{A}) \frac{tp \log d}{\varepsilon})$.*

---

**Algorithm 4** RobustPCA($\{X_i\}_{i \in [n]}, \epsilon, t$)

---

1: **Input:** $\{X_i\}_{i \in [n]}$ $\epsilon = O(\frac{1}{\log d \log \log d})$, $t \in [d]$ with $\Sigma_{t+1} \leq (1 - \gamma)\Sigma$ for $\gamma$ in Theorem 2
2: $w \leftarrow$ BoxedSchattenPacking (Proposition 2) on $\{\mathbf{A}_i = X_i X_i^\top\}_{i \in [n]}$, $\alpha \leftarrow \epsilon$, $p$ as in (5)
3: $\mathbf{M} = \sum_{i \in [n]} w_i X_i X_i^\top$
4: $\{z_j\}_{j \in [t]} = $ Power($t, \epsilon, p, \mathbf{M}$)
5: $\alpha_j \leftarrow$ 1DRobustVariance($\{X_i\}_{i \in [n]}, \mathbf{M}^{\frac{p-1}{2}} z_j / \|\mathbf{M}^{\frac{p-1}{2}} z_j\|_2, \epsilon$) for all $j \in [t]$
6: **return** $z_{j^*}$ for $j^* = \text{argmax}_{j \in [t]} \alpha_j$

---

Algorithm 4 is computationally bottlenecked by the application of Proposition 2 on Line 2 and the $t$ calls to 1DRobustVariance on Line 5, from which the runtime guarantee of Theorem 2 follows straightforwardly. To demonstrate correctness, we first certify the quality of the solution to (5).

**Lemma 5.** *Let $n = \Omega\left(\frac{d + \log \delta^{-1}}{(\epsilon \log \epsilon^{-1})^2}\right)$. With probability $1 - \frac{\delta}{2}$, the uniform distribution over $G$ attains value $(1 + \frac{\tilde{\epsilon}}{2})\|\Sigma\|_p$ for objective (5), where $\tilde{\epsilon} = C'\epsilon \log \epsilon^{-1}$ for a universal constant $C' > 0$.*

The proof of this is similar to results in e.g. [DKK$^+$19, Li18], and combines concentration guarantees with a union bound over all possible corruption sets $B$. This implies the following immediately, upon applying the guarantees of Proposition 2.

**Corollary 1.** *Let $w$ be the output of Line 2 of* RobustPCA. *Then, we have $\|w\|_\infty \leq \frac{1}{(1-2\epsilon)n}$, and $\left\|\sum_{i \in [n]} w_i X_i X_i^\top\right\|_p \leq (1 + \tilde{\epsilon}) \|\Sigma\|_p$ under the guarantee of Lemma 5.*

Let $w$ be the output of the solver. Recall that $\mathbf{M} = \sum_{i=1}^n w_i X_i X_i^\top$. Additionally, define

$$\mathbf{M}_G := \sum_{i \in G} w_i X_i X_i^\top, \ w_G := \sum_{i \in G} w_i, \ \mathbf{M}_B := \sum_{i \in B} w_i X_i X_i^\top, \ w_B := \sum_{i \in G} w_i . \tag{7}$$

Notice in particular that $\mathbf{M} = \mathbf{M}_G + \mathbf{M}_B$, and that all these matrices are PSD. We next prove the second, crucial fact, which says that $\mathbf{M}_G$ is a good approximator to $\Sigma$ in Loewner ordering:

**Lemma 6.** *Let $n = \Omega\left(\frac{d + \log \delta^{-1}}{(\epsilon \log \epsilon^{-1})^2}\right)$. With probability at least $1 - \frac{\delta}{2}$, $(1 + \tilde{\epsilon})\Sigma \succeq \mathbf{M}_G \succeq (1 - \tilde{\epsilon})\Sigma$.*

The proof combines the strategy in Lemma 5 with the SDP solver guarantee. Perhaps surprisingly, Corollary 1 and Lemma 6 are the only two properties about $\mathbf{M}$ that our final analysis of Theorem 2 will need. In particular, we have the following key geometric proposition, which carefully combines trace inequalities to argue that the corrupted points cannot create too many new large eigendirections.

**Proposition 4.** *Let $\mathbf{M} = \mathbf{M}_G + \mathbf{M}_B$ be so that $\|\mathbf{M}\|_p \leq (1 + \tilde{\epsilon}) \|\Sigma\|_p$, $\mathbf{M}_G \succeq 0$ and $\mathbf{M}_B \succeq 0$, and so that $(1 + \tilde{\epsilon})\Sigma \succeq \mathbf{M}_G \succeq (1 - \tilde{\epsilon})\Sigma$. Following notation of Algorithm 4, let*

$$\mathbf{M} = \sum_{j \in [d]} \lambda_j v_j v_j^\top, \ \Sigma = \sum_{j \in [d]} \sigma_j u_j u_j^\top \tag{8}$$

*be sorted eigendecompositions of $\mathbf{M}$ and $\Sigma$, so $\lambda_1 \geq \ldots \geq \lambda_d$, and $\sigma_1 \geq \ldots \geq \sigma_d$. Let $\gamma$ be as in Theorem 2, and assume $\sigma_{t+1} < (1 - \gamma)\sigma_1$. Then,*

$$\max_{j \in [t]} v_j^\top \Sigma v_j \geq (1 - \gamma) \|\Sigma\|_\infty .$$

With Proposition 4 in place, the recovery bound of Theorem 2 follows from an exact SVD. We show in Appendix D that the method is robust to approximations of the form (6), yielding our final claim.

## Broader Impact

Our work provides frameworks for learning properties about the covariance of sub-Gaussian distributions which have been corrupted under noise. As a key subroutine, we develop solvers for smoothed positive linear and semidefinite programs. We believe these results are interesting from an academic perspective, e.g. our techniques may be applicable generally for robust statistics and convex optimization researchers. Moreover, because our primary results concern robustness of models to arbitrarily corrupted data, we believe our methods may have practical implications for downstream tasks where protection against a malicious adversary is warranted. Similarly, as our main subroutine is a solver attaining strong computational guarantees for a wider variety of objectives than was previously known, it is possible that our methods can be leveraged to broaden the types of downstream tasks that can be performed. Namely, as $\ell_p$ norm packing linear program solvers have found applications in fair resource allocation, our hope is that our smoothed and mixed-norm guarantee semidefinite solvers can find similar applications in learning algorithms for objectives designed with fairness or privacy in mind.

## Footnotes

[1] See Section 2 for a formal definition.

[2] Throughout we use $\|\mathbf{M}\|_p$ to denote the Schatten $p$-norm (cf. Section 2 for more details).

[3]We say $g = \tilde{O}(f)$ if $g = O(f \log^c f)$ for some constant $c > 0$.

[4]In concurrent and independent work, [CMY20] develops width-independent solvers for Ky-Fan packing objectives, a different notion of generalization than the Schatten packing objectives we consider.

[5]We follow the distribution and corruption model described in Assumption 1.

[6]Corollary 4 gives a slightly stronger guarantee that reusing samples does not break dependencies of $u$.

[7]The [MRWZ16] solver also generalizes to covering and mixed objectives; we focus on packing in this work.

[8]Packing linear programs are sometimes expressed as the optimization problem $\max_{x \geq 0, \mathbf{A}x \leq \mathbf{1}} \|x\|_1$, similarly to (1); these problems are equivalent up to a standard binary search, see e.g. discussion in [JLL$^+$20].

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
