[Supplementary Material]

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

# A   Concentration

## A.1   Sub-Gaussian concentration

We use the following concentration facts on sub-Gaussian distributions following from standard techniques, and give an application bounding Schatten-norm deviations.

**Lemma 7.** *Under Assumption 1, there are universal constants $C_1$, $C_2$ such that*

$$\Pr\left[\sup_{\substack{v \in \mathbb{R}^d \\ \|v\|_2 = 1}} \left| v^\top \left( \frac{1}{n} \sum_{i \in G'} X_i X_i^\top - \mathbf{\Sigma} \right) v \right| - t v^\top \mathbf{\Sigma} v > 0 \right] \leq \exp\left( C_1 d - C_2 n \min(t, t^2) \right).$$

*Proof.* By observing (3), it is clear that the random vector $\widetilde{X} = \mathbf{\Sigma}^{-\frac{1}{2}} X$ for $X \sim \mathcal{D}$ has covariance $\mathbf{I}$ and sub-Gaussian proxy $c\mathbf{I}$. For any fixed unit vector $u$, by Lemma 1.12 of [RH17], the random variable $(u^\top \widetilde{X})^2 - 1$ is sub-exponential with parameter $16c$, so by Bernstein's inequality (Theorem 1.13, [RH17]), defining $\widetilde{X}_i = \mathbf{\Sigma}^{-\frac{1}{2}} X_i$ for each $X_i \sim \mathcal{D}$,

$$\Pr\left[ \left| u^\top \left( \frac{1}{n} \sum_{i \in G'} \widetilde{X}_i \widetilde{X}_i^\top - \mathbf{I} \right) u \right| > \frac{t}{2} \right] \leq \exp\left( -\frac{n}{2^{11} c^2} \min(t, t^2) \right).$$

For shorthand define $\mathbf{M} := \frac{1}{n} \sum_{i \in G'} \widetilde{X}_i \widetilde{X}_i^\top$, and let $\mathcal{N}$ be a maximal $\frac{1}{4}$-net of the unit ball (as measured in $\ell_2$ distance). By Lemma 1.18 of [RH17], $|\mathcal{N}| \leq 12^d$, so by a union bound,

$$\Pr\left[ \sup_{u \in \mathcal{N}} \left| u^\top (\mathbf{M} - \mathbf{I}) u \right| > \frac{t}{2} \right] \leq \exp\left( 3d - \frac{n}{2^{11} c^2} \min(t, t^2) \right).$$

Next, by a standard application of the triangle inequality (see e.g. Exercise 4.3.3, [Ver16])

$$\sup_{\substack{v \in \mathbb{R}^d \\ \|v\|_2 = 1}} \left| v^\top (\mathbf{M} - \mathbf{I}) v \right| \leq 2 \sup_{u \in \mathcal{N}} \left| u^\top (\mathbf{M} - \mathbf{I}) u \right| \leq t$$

with probability at least $1 - \exp\left( C_1 d - C_2 n \min(t, t^2) \right)$ for appropriate $C_1$, $C_2$. The conclusion follows since its statement is scale invariant, so it suffices to show as we have that

$$\Pr\left[\sup_{\substack{v \in \mathbb{R}^d \\ \|v\|_{\mathbf{\Sigma}} = 1}} \left| v^\top \left( \frac{1}{n} \sum_{i \in G'} X_i X_i^\top - \mathbf{\Sigma} \right) v \right| - t v^\top \mathbf{\Sigma} v > 0 \right] \leq \exp\left( C_1 d - C_2 n \min(t, t^2) \right).$$

$\square$

**Corollary 2.** *Let $p \geq 2$. Under Assumption 1, there are universal constants $C_1$, $C_2$ with*

$$\Pr\left[ \left\| \frac{1}{n} \sum_{i \in G'} X_i X_i^\top - \mathbf{\Sigma} \right\|_p > t \|\mathbf{\Sigma}\|_p \right] \leq \exp\left( C_1 d - C_2 n \min(t, t^2) \right).$$

*Proof.* Suppose the event in Lemma 7 does not hold, which happens with probability at least $1 - \exp(C_1 d - C_2 n \min(t, t^2))$. Define for shorthand $\mathbf{M} := \frac{1}{n} \sum_{i \in G'} X_i X_i^\top - \mathbf{\Sigma}$ and let its spectral decomposition be $\sum_{j \in [d]} \lambda_j v_j v_j^\top$. By the triangle inequality and Fact 2,

$$\|\mathbf{M}\|_p \leq \sum_{j \in [d]} \frac{|\lambda_j|^{p-1}}{\|\mathbf{M}\|_p^{p-1}} \left| v_j^\top \left( \frac{1}{n} \sum_{i \in G'} X_i X_i^\top - \mathbf{\Sigma} \right) v_j \right|$$

$$\leq t \sum_{j \in [d]} \frac{|\lambda_j|^{p-1}}{\|\mathbf{M}\|_p^{p-1}} v_j^\top \mathbf{\Sigma} v_j = t \left\langle \sum_{j \in [d]} \frac{|\lambda_j|^{p-1}}{\|\mathbf{M}\|_p^{p-1}} v_j v_j^\top, \mathbf{\Sigma} \right\rangle \leq t \|\mathbf{\Sigma}\|_p.$$

In the last inequality, we used that $\sum_{j \in [d]} \frac{|\lambda_j|^{p-1}}{\|\mathbf{M}\|_p^{p-1}} v_j v_j^\top$ has unit $\ell_q$ norm, and applied Fact 2.   $\square$

## A.2 Concentration under weightings in $\mathfrak{S}_\epsilon^n$

We consider concentration of the empirical covariance under weightings which are not far from uniform, in spectral and Schatten senses.

**Lemma 8.** *Under Assumption 1, let $\delta \in [0, 1]$, $p \geq 2$, and $n = \Omega\left(\frac{d + \log \delta^{-1}}{(\epsilon \log \epsilon^{-1})^2}\right)$ for a sufficiently large constant. Then for a universal constant $C_3$,*

$$\Pr\left[\exists w \in \mathfrak{S}_\epsilon^n \ \middle| \ \left\|\sum_{i \in G'} w_i X_i X_i^\top - \Sigma\right\|_p > C_3 \cdot \epsilon \log \epsilon^{-1} \left\|\Sigma\right\|_p\right] \leq \frac{\delta}{2}.$$

*Proof.* Because the vertices of $\mathfrak{S}_\epsilon^n$ are uniform over sets $S \subseteq G'$ with $|S| = (1 - \epsilon)n$ (see e.g. Section 4.1, [DKK$^+$19]), by convexity of the Schatten-$p$ norm it suffices to prove

$$\Pr\left[\exists S \text{ with } |S| = (1 - \epsilon)n \ \middle| \ \left\|\frac{1}{(1-\epsilon)n}\sum_{i \in S} X_i X_i^\top - \Sigma\right\|_p > C_3 \cdot \epsilon \log \epsilon^{-1} \left\|\Sigma\right\|_p\right] \leq \frac{\delta}{4}.$$

For any fixed $S$, and recalling $|S^c| = \epsilon n$, we can decompose this sum as

$$\frac{1}{(1-\epsilon)n}\sum_{i \in S} X_i X_i^\top = \frac{1}{1-\epsilon}\left(\frac{1}{n}\sum_{i \in G'} X_i X_i^\top\right) - \frac{\epsilon}{1-\epsilon}\left(\frac{1}{|S^c|}\sum_{i \in S^c} X_i X_i^\top\right). \tag{9}$$

By applying Corollary 2, it follows that by setting $t = \frac{1-\epsilon}{2} \cdot \epsilon \log \epsilon^{-1}$ and our choice of $n$ that

$$\Pr\left[\left\|\frac{1}{n}\sum_{i \in G'} X_i X_i^\top - \Sigma\right\|_p > \frac{1-\epsilon}{2} \cdot \epsilon \log \epsilon^{-1} \left\|\Sigma\right\|_p\right] \leq \frac{\delta}{4}. \tag{10}$$

Moreover, for any fixed $S^c$, setting $t = \frac{1-\epsilon}{2} \cdot C_3 \log \epsilon^{-1}$ where $C_3$ is a sufficiently large constant, so that for sufficiently small $\epsilon$, $t = \min(t, t^2)$,

$$\Pr\left[\left\|\frac{1}{\epsilon n}\sum_{i \in S^c} X_i X_i^\top - \Sigma\right\|_p > \frac{1-\epsilon}{2} \cdot C_3 \cdot \log \epsilon^{-1} \left\|\Sigma\right\|_p\right] \leq \exp\left(C_1 d - C_2 \epsilon n t\right)$$

$$\leq \exp\left(-\left(\log \delta^{-1} + n\epsilon \log \epsilon^{-1}\right)\right)$$

$$\leq \frac{\delta}{4\binom{n}{\epsilon n}}. \tag{11}$$

Here, we used that $\log\binom{n}{\epsilon n} = O\left(n\epsilon \log \epsilon^{-1}\right)$. Finally, union bounding over all possible sets $S^c$ imply that with probability at least $1 - \frac{\delta}{2}$, the following events hold:

$$\left\|\frac{1}{n}\sum_{i \in G'} X_i X_i^\top - \Sigma\right\|_p < \frac{1-\epsilon}{2} \cdot \epsilon \log \epsilon^{-1} \left\|\Sigma\right\|_p,$$

$$\left\|\frac{1}{|S^c|}\sum_{i \in S^c} X_i X_i^\top - \Sigma\right\|_p < \frac{1-\epsilon}{2} \cdot C_3 \cdot \log \epsilon^{-1} \left\|\Sigma\right\|_p \text{ for all } S \text{ with } |S| = (1 - \epsilon)n.$$

Combining these bounds in the context of (9) after applying the triangle inequality, we have with probability at least $1 - \frac{\delta}{2}$ for all $S$ the desired conclusion,

$$\left\|\frac{1}{(1-\epsilon)n}\sum_{i \in S} X_i X_i^\top - \Sigma\right\|_p < C_3 \cdot \epsilon \log \epsilon^{-1} \left\|\Sigma\right\|_p.$$

$\square$

**Corollary 3.** *Under Assumption 1, let $n = \Omega\left(\frac{d + \log \delta^{-1}}{(\epsilon \log \epsilon^{-1})^2}\right)$ for a sufficiently large constant. For universal $C_3$ and all $w \in \mathfrak{S}_\epsilon^n$, with probability at least $1 - \frac{\delta}{2}$,*

$$C_3 \cdot \epsilon \log \epsilon^{-1} \mathbf{\Sigma} \succeq \sum_{i \in G'} w_i X_i X_i^\top - \mathbf{\Sigma} \succeq -C_3 \cdot \epsilon \log \epsilon^{-1} \mathbf{\Sigma}.$$

*Proof.* Consider any unit vector $v \in \mathbb{R}^d$. By similar arguments as in (10), (11), and applying a union bound over all $S$ with $|S| = (1 - \epsilon)n$, with probability at least $1 - \frac{\delta}{2}$, it follows from Lemma 7 that

$$\left| v^\top \left( \frac{1}{n} \sum_{i \in G'} X_i X_i^\top - \mathbf{\Sigma} \right) v \right| < \frac{1 - \epsilon}{2} \cdot \epsilon \log \epsilon^{-1} v^\top \mathbf{\Sigma} v, \tag{12}$$

$$\left| v^\top \left( \frac{1}{|S^c|} \sum_{i \in S^c} X_i X_i^\top - \mathbf{\Sigma} \right) v \right| < \frac{1 - \epsilon}{2} \cdot C_3 \cdot \log \epsilon^{-1} v^\top \mathbf{\Sigma} v . \tag{13}$$

Therefore, again using the formula (9) and the triangle inequality yields the desired conclusion for all directions $v$, which is equivalent to the spectral bound of the lemma statement. $\qquad\square$

# B  Deferred proofs from Section 3

## B.1  Robust univariate variance estimation

In this section, we prove Lemma 1, which allows us to robustly estimate the quadratic form of a vector in the covariance of a sub-Gaussian distribution from corrupted samples. Algorithm 5 is folklore, and intuitively very simple; it projects all samples onto $u$, throws away the $2\epsilon$ fraction of points with largest magnitude in this direction, and takes the mean of the remaining set.

---

**Algorithm 5** Univariate variance estimation: 1DRobustVariance($\{X_i\}_{i \in [n]}, \epsilon, u$)

---

**Input:** $\{X_i\}_{i \in [n]}$, $\epsilon > 0$, and a unit vector $u$
Let $a_i = \langle X_i, u \rangle^2$ for $i = 1, \dots, n$
Sort the $a_i$ in increasing order. WLOG assume $a_1 \leq a_2 \leq \dots \leq a_n$.
**return** $\sigma_u^2 = \frac{1}{(1 - 2\epsilon)n} \sum_{i=1}^{(1 - 2\epsilon)n} a_i$

---

We require the following helper fact.

**Fact 3.** *Let $Z$ be a sub-exponential random variable with parameter at most $\lambda$[9], and let $\epsilon \in [0, 1]$. Then, for any event $E$ with $\Pr[Z \in E] \leq \epsilon$, $|\mathbb{E}[Z \cdot \mathbf{1}[Z \in E]]| \leq 2\lambda \epsilon \log \epsilon^{-1}$.*

*Proof.* We have by Hölder's inequality that for any $p, q \geq 1$ with $p^{-1} + q^{-1} = 1$,

$$|\mathbb{E}[Z \cdot \mathbf{1}[Z \in E]]| \leq \mathbb{E}[|Z|^p]^{1/p} \cdot \epsilon^{1/q} \leq 2\lambda p \cdot \epsilon^{1/q}.$$

The second inequality is Lemma 1.10 [RH17]. Setting $p = \log \epsilon^{-1}$ yields the result. $\qquad\square$

**Lemma 1.** *Under Assumption 1, let $\delta \in [0, 1]$, $n = \Omega\left(\frac{\log \delta^{-1}}{(\epsilon \log \epsilon^{-1})^2}\right)$, and $u \in \mathbb{R}^d$ be a fixed unit vector. Algorithm 5, 1DRobustVariance, takes input $\{X_i\}_{i \in [n]}$, $u$, and $\epsilon$, and outputs $\sigma_u^2$ with $|u^\top \mathbf{\Sigma} u - \sigma_u^2| < C u^\top \mathbf{\Sigma} u \cdot \epsilon \log \epsilon^{-1}$ with probability at least $1 - \delta$, and runs in time $O(nd + n \log n)$, for $C$ a fixed multiple of the parameter $c$ in Assumption 1.*

*Proof.* The runtime claim is immediate; we now turn our attention to correctness. We follow notation of Assumption 1, and in a slight abuse of notation, also define $a_i = \langle X_i, u \rangle^2$ for $i \in G'$. First, for $X \sim \mathcal{D}$, then $\langle u, X \rangle^2 - u^\top \mathbf{\Sigma} u$ is sub-exponential with parameter at most $16c u^\top \mathbf{\Sigma} u$ (Lemma 1.12, [RH17]). By Bernstein's inequality, we have that if $X \sim \mathcal{D}$, then for all $t \geq 1$,

$$\Pr\left[\langle X, u \rangle^2 > 32ct u^\top \mathbf{\Sigma} u\right] \leq \exp(-t) . \tag{14}$$

Using this in a standard Chernoff bound, we have that with probability $1 - \frac{\delta}{2}$,

$$\frac{\left|\{i \in G' : a_i > 64c \log \epsilon^{-1} \cdot u^\top \Sigma u\}\right|}{n} \leq \epsilon . \tag{15}$$

Let $T = 64c \log \epsilon^{-1} \cdot u^\top \Sigma u$, and let $Y$ be distributed as $(\langle u, X\rangle^2 - u^\top \Sigma u) \cdot \mathbf{1}[\langle u, X\rangle^2 \leq T]$, where $X \sim \mathcal{D}$. We observe $Y - \mathbb{E}[Y]$ is also sub-exponential with parameter $16cu^\top \Sigma u$, and that by Fact 3,

$$|\mathbb{E}[Y]| \leq 32cu^\top \Sigma u \epsilon \log \epsilon^{-1}. \tag{16}$$

Define the interval $I = [0, T]$ and let $S$ be the set of points in $[n]$ that survive the truncation procedure, so that $\sigma_u^2 = \frac{1}{|S|} \sum_{i \in S} a_i$. Given event (15), $a_i \in I$ for all $i \in S$, since there are at most $\epsilon n$ points in $G$ outside $I$, and $|B| \leq \epsilon n$. We decompose the deviation as follows:

$$\begin{aligned}
\sum_{i \in S} a_i - |S| u^\top \Sigma u &= \sum_{i \in G \cap S} (a_i - u^\top \Sigma u) + \sum_{i \in B \cap S} (a_i - u^\top \Sigma u) \\
&= \sum_{i \in G' \cap I} (a_i - u^\top \Sigma u) + \sum_{i \in B \cap S} (a_i - u^\top \Sigma u) \\
&\quad - \sum_{i \in (G' \setminus G) \cap I} (a_i - u^\top \Sigma u) - \sum_{i \in (G \cap I) \setminus S} (a_i - u^\top \Sigma u).
\end{aligned} \tag{17}$$

Here we overloaded $i \in I$ to mean that $a_i$ lies in the interval $I$, and conditioned on $S$ lying entirely in $I$. We bound each of these terms individually. First, for all $i \in G' \cap I$, conditioning on (15) (i.e. all $a_i \in I$), $a_i - u^\top \Sigma u$ is an independent sample from $Y$. Thus, by (16) and Bernstein's inequality,

$$\begin{aligned}
\left| \frac{1}{|G' \cap I|} \sum_{i \in G' \cap I} (a_i - u^\top \Sigma u) \right| &\leq \left| \frac{1}{|G' \cap I|} \sum_{i \in G' \cap I} (a_i - u^\top \Sigma u) - \mathbb{E}[Y] \right| + 32cu^\top \Sigma u \epsilon \log \epsilon^{-1} \\
&\leq 64c \cdot u^\top \Sigma u \epsilon \log \epsilon^{-1},
\end{aligned} \tag{18}$$

with (conditional) probability at least $1 - \frac{\delta}{2}$. By a union bound, both events occur with probability at least $1 - \delta$; condition on this for the remainder of the proof. Under this assumption, we control the other three terms of (17). Observe that $|B \cap S| \leq \epsilon n$, $|(G' \setminus G) \cap I| \leq \epsilon n$, and $|(G \cap I) \setminus S| \leq \epsilon n$. Further, by definition of $I$, every summand is at most $64c \log \epsilon^{-1} \cdot u^\top \Sigma u$. Thus,

$$\left| \sum_{i \in B \cap S} (a_i - u^\top \Sigma u) \right| \leq 64c\epsilon n \log \epsilon^{-1} \cdot u^\top \Sigma u, \tag{19}$$

$$\left| \sum_{i \in (G' \setminus G) \cap I} (a_i - u^\top \Sigma u) \right| \leq 64c\epsilon n \log \epsilon^{-1} \cdot u^\top \Sigma u. \tag{20}$$

$$\left| \sum_{i \in (G' \cap I) \setminus S} (a_i - u^\top \Sigma u) \right| \leq 64c\epsilon n \log \epsilon^{-1} \cdot u^\top \Sigma u. \tag{21}$$

Combining (18), (19), (20), and (21) in derivation (17) and dividing by $|S|$ yields the claim. $\qquad\square$

Finally, we also give an alternative set of conditions under which we can certify correctness of 1DRobustVariance. Specifically, this assumption will be useful in lifting indpendence assumptions between $u$ and our samples $\{X_i\}_{i \in [n]}$ in repeated calls within Algorithm 6.

**Assumption 2.** *Under Assumption 1, let the following conditions hold for universal constant $C_4$:*

$$C_4 \epsilon \log \epsilon^{-1} \cdot \Sigma \succeq \frac{1}{n} \sum_{i \in G'} X_i X_i^\top - \Sigma \succeq -C_4 \epsilon \log \epsilon^{-1} \cdot \Sigma, \tag{22}$$

$$C_4 \log \epsilon^{-1} \cdot \Sigma \succeq \sum_{i \in G'} w_i \left( X_i X_i^\top - \Sigma \right) \succeq -C_4 \log \epsilon^{-1} \cdot \Sigma \text{ for all } w \in \mathfrak{S}_{1-\epsilon}^n. \tag{23}$$

Note that (23) is a factor $\epsilon$ weaker in its guarantee than Corollary 3, and is over weights in a different set $\mathfrak{S}_{1-\epsilon}^n$. Standard sub-Gaussian concentration (i.e. an unweighted variant of Corollary 3) and modifying the proof of Corollary 3 to take the constraint set $\mathfrak{S}_{1-\epsilon}^n$ and normalizing over vertex sets of size $\epsilon n$ yield the following conclusion.

**Lemma 9.** *Let $n = \Omega\left(\frac{d + \log \delta^{-1}}{(\epsilon \log \epsilon^{-1})^2}\right)$ for a sufficiently large constant. Assumption 2 holds with probability at least $1 - \frac{\delta}{2}$.*

We give a variant of Lemma 1 with slightly stronger guarantees for 1DRobustVariance; specifically, it holds for all $u$ simultaneously for a fixed set of samples satisfying Assumption 2.

**Corollary 4.** *Under Assumption 2, Algorithm 5 outputs $\sigma_u^2$ with $|u^\top \Sigma u - \sigma_u^2| < C u^\top \Sigma u \cdot \epsilon \log \epsilon^{-1}$, for $C$ a fixed multiple of the parameter $c$ in Assumption 1, and runs in time $O(nd + n \log n)$.*

*Proof.* We discuss how to modify the derivations from Lemma 1 appropriately in the absence of applications of Bernstein's inequality. First, note that appropriately combining (22) and (23) in a derivation such as (9) yields the following bound (deterministically under Assumption 2):

$$C_4 \epsilon \log \epsilon^{-1} \cdot \Sigma \succeq \sum_{i \in G'} w_i \left(X_i X_i^\top - \Sigma\right) \succeq -C_4 \epsilon \log \epsilon^{-1} \Sigma \text{ for all } w \in \mathfrak{S}_{3\epsilon}^n. \tag{24}$$

Now, consider the decomposition (17). We claim first that similarly to (19), (20), (21) we can bound each summand in the latter three terms by $O(u^\top \Sigma u \log \epsilon^{-1})$; to prove this, it suffices to show that at least one filtered $a_i$ attains this bound, as then by definition of the algorithm, each non-filtered $a_i$ will as well. Note that a fraction between $\epsilon$ and $2\epsilon$ of points in $G \subset G'$ is filtered (since there are only $\epsilon n$ points from $B$). The assumption (23) then implies precisely the desired bound on some filtered $a_i$ by placing uniform mass on filtered points from $G$, and applying pigeonhole. So, all non-filtered $a_i$ are bounded by $O(u^\top \Sigma u \log \epsilon^{-1})$, yielding analogous statements to (19), (20), (21).

Finally, an analogous derivation to (18) follows via an application of the bound (24), where we place uniform mass on the set $G' \cap I$ and adjust constants appropriately, since the above argument shows that under the assumption (23), we have that at most $2\epsilon n$ indices $i \in G'$ have $a_i \notin I$. $\square$

## B.2 Preliminaries

For convenience, we give the following preliminaries before embarking on our proof of Theorem 1 and giving guarantees on Algorithm 6. First, we state a set of assumptions which augments Assumption 2 with one additional condition, used in bounding the iteration count of our algorithm.

**Assumption 3.** *Under Assumption 1, let Assumption 2 hold, as well as the following additional condition for the same universal constant $C_4$:*

$$\|X_i\|_2^2 \leq C_4 \log \frac{n}{\delta} \cdot \mathrm{Tr}(\Sigma) \text{ for all } i \in G. \tag{25}$$

Standard sub-Gaussian concentration inequalities and a union bound, combined with our earlier claim Lemma 9, then yield the following guarantee.

**Lemma 10.** *Let $n = \Omega\left(\frac{d + \log \delta^{-1}}{(\epsilon \log \epsilon^{-1})^2}\right)$ for a sufficiently large constant. Assumption 3 holds with probability at least $1 - \delta$.*

## B.3 Analysis of PCAFilter

For this section, for any nonnegative weights $w$, define $\mathbf{M}(w) := \sum_{i \in [n]} w_i X_i X_i^\top$. We now state our algorithm, PCAFilter. At all iterations $t$, it maintains a current nonnegative weight vector $w^{(t)}$ (initialized to be the uniform distribution on $[n]$), preserving the following invariants for all $t$:

$$w_i^{(t-1)} \geq w_i^{(t)} \text{ for all } i \in [n], \ \sum_{i \in B} w_i^{(t-1)} - w_i^{(t)} \geq \sum_{i \in G} w_i^{(t-1)} - w_i^{(t)}. \tag{26}$$

We now state our method as Algorithm 6; note that the update to $w^{(t)}$ is of the form in Lemma 2.

**Algorithm 6** PCAFilter($\{X_i\}_{i \in [n]}, \epsilon$)

1: Remove all points $i \in [n]$ with $\|X_i\|_2^2 > c \log(\frac{n}{\delta}) \cdot \text{Tr}(\mathbf{\Sigma})$
2: $w_i^{(0)} \leftarrow \frac{1}{n}$ for all $i \in [n]$, $t \leftarrow 1$
3: $u_1 \leftarrow$ approximate top eigenvector of $\mathbf{M}(w^{(0)})$
4: $\sigma_1^2 \leftarrow$ 1DRobustVariance($\{X_i\}_{i \in [n]}, \epsilon, u_1$)
5: **while** $u_t^\top \mathbf{M}(w^{(t-1)}) u_t > (1 + 5C_5 \epsilon \log \epsilon^{-1}) \sigma_t^2$, where $C_5 = \max(C, C_4)$ from constants in Assumption 2, Corollary 4 **do**
6:     $a_i \leftarrow \langle u_t, X_i \rangle^2$ for all $i \in [n]$
7:     Sort (permute) the indices $[n]$ so the $a_i$ are in increasing order (with $a_1$ smallest, $a_n$ largest)
8:     Let $\ell$ be the largest index with $\sum_{i=\ell}^n w_i \geq 2\epsilon$
9:     Define
$$w_i^{(t)} \leftarrow \begin{cases} \left(1 - \frac{a_i}{a_n}\right) w_i^{(t-1)} & \ell \leq i \leq n \\ w_i^{(t-1)} & i < \ell \end{cases}$$

10:     $u_t \leftarrow$ approximate top eigenvector of $\mathbf{M}(w^{(t)})$
11:     $\sigma_t^2 \leftarrow$ 1DRobustVariance($\{X_i\}_{i \in [n]}, u_t, \epsilon$)
12:     $t \leftarrow t + 1$
13: **end while**
14: **return** $u_t$

We assume that in Line 8, we also have $\sum_{i=\ell}^n w_i \leq 3\epsilon$, as we can assume at least one point is corrupted i.e. $\epsilon \geq \frac{1}{n}$ (else standard algorithms suffice for our setting), so adding an additional $w_i$ can only change the sum by $\epsilon$. We first prove invariants (26) are preserved; at a high level, we simply demonstrate that Lemma 2 holds via concentration on $G$ and lack of termination.

**Lemma 11.** *Under Assumption 2, for any iteration $t$ of Algorithm 6, suppose* (26) *held for all iterations $t' \leq t - 1$. Then,* (26) *holds at iteration $t$.*

*Proof.* The first part of (26) is immediate by observing the update in Line 9, so we show the second. We drop subscripts and superscripts for conciseness and focus on a single iteration $t$. Let $I_B = \{\ell, \dots, n\} \cap B$, and $I_G = \{\ell, \dots, n\} \cap G$. By Lemma 2, it suffices to demonstrate that

$$\sum_{i \in I_B} w_i a_i > \sum_{i \in I_G} w_i a_i. \tag{27}$$

First, $\sum_{i \in I_B} w_i \leq \epsilon$, so by definition of index $\ell$, we have $\epsilon \leq \sum_{i \in I_G} w_i \leq 2\epsilon$. Define $\widetilde{w}_i = \frac{w_i}{\sum_{i \in I_G} w_i}$ if $i \in I_G$, and 0 otherwise, and observe $\widetilde{w} \in \mathfrak{S}_{1-2\epsilon}^n$. By modifying constants appropriately from (23), it follows from definition of $a_i = u^\top X_i X_i^\top u$ that

$$\sum_{i \in I_G} w_i a_i \leq \left( \sum_{i \in I_G} w_i \right) \cdot C_4 \log \epsilon^{-1} \cdot u^\top \mathbf{\Sigma} u \leq 2C_4 \epsilon \log \epsilon^{-1} \cdot u^\top \mathbf{\Sigma} u. \tag{28}$$

On the other hand, by (24) we know that the total quadratic form over $G$ is bounded as

$$\sum_{i \in G} w_i a_i < \left( \sum_{i \in G} w_i \right) \left( 1 + C_4 \epsilon \log \epsilon^{-1} \right) u^\top \mathbf{\Sigma} u < \left( 1 + C_4 \epsilon \log \epsilon^{-1} \right) u^\top \mathbf{\Sigma} u. \tag{29}$$

Here, we applied the observation that the normalized $w_i$ restricted to $G$ are in $\mathfrak{S}_{1-3\epsilon}^n$ (e.g. using Lemma 12 inductively). However, since we did not terminate (Line 5), we must have by $u_t$ being a top eigenvector and Corollary 4 (we defer discussions of inexactness to Theorem 1) that

$$\sum_{i \in [n]} w_i a_i \geq (1 + 5C_5 \epsilon \log \epsilon^{-1}) \sigma_t^2 \geq (1 + 4C_4 \epsilon \log \epsilon^{-1}) \cdot u^\top \mathbf{\Sigma} u$$

$$\implies \sum_{i \in B} w_i a_i > 3C_4 \epsilon \log \epsilon^{-1} \cdot u^\top \mathbf{\Sigma} u.$$

To obtain the last conclusion, we used (29). Finally, note that for all $i \in B \setminus I_B$,

$$a_i \le a_\ell \le \sum_{i \in I_G} \widetilde{w}_i a_i \le C_4 \log \epsilon^{-1} \cdot u^\top \mathbf{\Sigma} u$$

by rearranging (28). This implies that

$$\sum_{i \in B \setminus I_B} w_i a_i \le \left( \sum_{i \in B \setminus I_B} w_i \right) \cdot C_4 \log \epsilon^{-1} \cdot u^\top \mathbf{\Sigma} u \le C_4 \epsilon \log \epsilon^{-1} \cdot u^\top \mathbf{\Sigma} u.$$

Thus, the desired inequality (27) follows from combining the above derivations, e.g. using (28) and

$$\sum_{i \in I_B} w_i a_i = \sum_{i \in B} w_i a_i - \sum_{i \in B \setminus I_B} w_i a_i > 2 C_4 \epsilon \log \epsilon^{-1} \cdot u^\top \mathbf{\Sigma} u.$$

$\square$

Lemma 11 yields for all $t$ that $\sum_{i \in B} w_i^{(0)} - w_i^{(t)} \ge \sum_{i \in G} w_i^{(0)} - w_i^{(t)}$ by telescoping. Note that we can only remove at most $2\epsilon$ mass from $w$ total, as $\sum_{i \in B} w_i^{(0)} - w_i^{(t)} \le \epsilon$. Denote for shorthand normalized weights $v^{(t)} := \frac{w^{(t)}}{\|w^{(t)}\|_1}$. Then, the following is immediate by $\|w^{(t)}\|_1 \ge 1 - 2\epsilon$.

**Lemma 12.** *Under Assumption 2, in all iterations $t$ of Algorithm 6, $v^{(t)} \in \mathfrak{S}_{2\epsilon}^n$.*

Using Lemma 12, we show that the output has the desired quality of being a large eigenvector.

**Lemma 13.** *Under Assumption 2, let the output of Algorithm 6 be $u_T$. Then for a universal constant $C^\star$, $u_T^\top \mathbf{\Sigma} u_T \ge (1 - C^\star \epsilon \log \epsilon^{-1}) \|\mathbf{\Sigma}\|_\infty$.*

*Proof.* We assume for now that $u_T$ is an exact top eigenvector, and discuss inexactness while proving Theorem 1. By (24) and Lemma 12, as then the normalized restriction of $w^{(T)}$ to $G$ is in $\mathfrak{S}_{3\epsilon}^n$,

$$\mathbf{M}(w^{(T)}) \succeq \sum_{i \in G} w_i^{(T)} X_i X_i^\top \succeq \left(1 - 2C_4 \epsilon \log \epsilon^{-1}\right) \mathbf{\Sigma}$$

$$\implies u_T^\top \mathbf{M}(w^{(T)}) u_T \ge \left(1 - 2C_4 \epsilon \log \epsilon^{-1}\right) \|\mathbf{\Sigma}\|_\infty.$$

We used the Courant-Fischer characterization of eigenvalues, and that $u_T$ is a top eigenvector of $\mathbf{M}(w^{(T)})$. Moreover, by termination conditions and Corollary 4 (correctness of 1DRobustVariance),

$$(1 + C\epsilon \log \epsilon^{-1}) u_T^\top \mathbf{\Sigma} u_T \ge \sigma_T^2 \ge (1 + 5C_5 \epsilon \log \epsilon^{-1})^{-1} u_T^\top \mathbf{M}(w^{(T)}) u_T.$$

Combining these two bounds and rescaling yields the conclusion. $\square$

Finally, we prove our main guarantee about Algorithm 6.

**Theorem 1.** *Under Assumption 1, let $\delta \in [0, 1]$, and $n = \Omega\left(\frac{d + \log \delta^{-1}}{(\epsilon \log \epsilon^{-1})^2}\right)$. Algorithm 6 runs in time $O(\frac{nd^2}{\epsilon} \log \frac{n}{\delta\epsilon} \log \frac{n}{\delta})$, and outputs $u$ with $u^\top \mathbf{\Sigma} u > (1 - C^\star \epsilon \log \epsilon^{-1}) \|\mathbf{\Sigma}\|_\infty$, for $C^\star$ a fixed multiple of parameter $c$ in Assumption 1, with probability at least $1 - \delta$.*

*Proof.* First, we will operate under Assumption 3, which holds with probability at least $1 - \delta$. It is clear that the analyses of Lemma 11 and 13 hold with $1 - \Theta(\epsilon \log \epsilon^{-1})$ multiplicative approximations of top eigenvector computation, which the power method approximates with high probability. Thus, each iteration takes time $O\left(\frac{nd}{\epsilon} \log \frac{n}{\delta\epsilon}\right)$, where we will union bound over the number of iterations.

We now give an iteration bound: in any iteration where we do not terminate, Lemma 2 implies

$$\sum_{i=1}^{n} w_i^{(t-1)} - w_i^{(t)} \geq \frac{1}{2 \max_{i \in [n]} \langle u_t, X_i \rangle^2} \sum_{i=\ell}^{n} w_i a_i$$

$$\geq \frac{1}{2 C_4 \log \frac{n}{\delta} \cdot \mathrm{Tr}(\boldsymbol{\Sigma})} \sum_{i=\ell}^{n} w_i a_i$$

$$\geq \frac{1}{2 C_4 \log \frac{n}{\delta} \cdot \mathrm{Tr}(\boldsymbol{\Sigma})} \left( \frac{\sum_{i=\ell}^{n} w_i}{\sum_{i \in [n]} w_i} \right) \sum_{i \in [n]} w_i a_i$$

$$= \Omega \left( \epsilon \cdot \frac{\|\boldsymbol{\Sigma}\|_{\infty}}{\log \frac{n}{\delta} \cdot \mathrm{Tr}(\boldsymbol{\Sigma})} \right) = \Omega \left( \frac{\epsilon}{d \log \frac{n}{\delta}} \right).$$

Here, the second line used Assumption 3, the third used that the $a_i$ are in sorted order, and the last used the definition of $\ell$ as well as the derivations of Lemma 13 (specifically, that $\mathbf{M}(w)$ spectrally dominates $(1 - O(\epsilon \log \epsilon^{-1}))\boldsymbol{\Sigma}$ for roughly uniform $w$). The conclusion follows since there can be at most $O(d \log \frac{n}{\delta})$ iterations, as the algorithm terminates when a $2\epsilon$ fraction of the mass is removed, giving the overall runtime claim. $\qquad\square$

## C   Deferred proofs from Section 4

### C.1   Proofs from Section 4.2

Since our notion of approximation is multiplicative, we can assume without more than constant loss that $\mathbf{A}$ has bounded entries. This observation is standard, and formalized in the following lemma.

**Lemma 14** (Entrywise bounds on $\mathbf{A}$). *Feasibility of Problem 2 is unaffected (up to constants in $\epsilon$) by removing columns of $\mathbf{A}$ with entries larger than $n\epsilon^{-1}$.*

*Proof.* If $\mathbf{A}_{ji} > n\epsilon^{-1}$ for any entry, then $x_i \leq \frac{\epsilon(1+\epsilon)}{n}$, else $\|\mathbf{A}x\|_p$ is already larger than $1 + \epsilon$. Ignoring all such entries of $x$ and rescaling can only change the objective by a $1 + O(\epsilon)$ factor. $\quad\square$

**Lemma 3.** *In all iterations $t$ of Algorithm 2, defining $\Phi_t := \|\mathbf{A}w_t\|_p - \|w_t\|_1$, $\Phi_{t+1} \leq \Phi_t$.*

*Proof.* Fix an iteration $t$. Define $\delta = \eta g_t$, and note $w_{t+1} = w_t + \delta \circ w_t$; henceforth in this proof, we will drop subscripts $t$ when clear. Observe that

$$\|\mathbf{A}w_{t+1}\|_p = \|\mathbf{A}((1 + \delta) \circ w)\|_p = \left( \sum_{j \in [d]} [\mathbf{A}w]_j^p \left( 1 + \frac{[\mathbf{A}(\delta \circ w_t)]_j}{[\mathbf{A}w_t]_j} \right)^p \right)^{1/p}.$$

As $g \leq \mathbf{1} \implies \delta \leq p^{-1}\mathbf{1}$, $\frac{\mathbf{A}(\delta \circ w_t)}{\mathbf{A}w_t} \leq p^{-1}$ entrywise. Via $(1 + x)^p \leq \exp(px) \leq 1 + px + p^2x^2$ for $x \leq p^{-1}$, it follows that

$$\|\mathbf{A}((1 + \delta) \circ w)\|_p \leq \left( \sum_{j \in [d]} [\mathbf{A}w]_j^p \left( 1 + \frac{p[\mathbf{A}(\delta \circ w)]_j}{[\mathbf{A}w]_j} + \left( \frac{p[\mathbf{A}(\delta \circ w)]_j}{[\mathbf{A}w]_j} \right)^2 \right) \right)^{1/p}.$$

By direct manipulation of the above quantity, and recalling we defined $v = \frac{\mathbf{A}w}{\|\mathbf{A}w\|_p}$,

$$\left( \sum_{j \in [d]} \left( [\mathbf{A}w]_j^p + p[\mathbf{A}w]_j^{p-1}[\mathbf{A}(\delta \circ w)]_j + p^2[\mathbf{A}w]_j^{p-2}[\mathbf{A}(\delta \circ w)]_j^2 \right) \right)^{1/p}$$

$$= \left( \|\mathbf{A}w\|_p^p \sum_{j \in [d]} \left( v_j^p + p v_j^{p-1}\frac{[\mathbf{A}(\delta \circ w)]_j}{\|\mathbf{A}w\|_p} + p^2 v_j^{p-2}\left( \frac{[\mathbf{A}(\delta \circ w)]_j}{\|\mathbf{A}w\|_p} \right)^2 \right) \right)^{1/p}$$

$$= \|\mathbf{A}w\|_p \left( 1 + \sum_{j \in [d]} \left( p v_j^{p-1}\frac{[\mathbf{A}(\delta \circ w)]_j}{\|\mathbf{A}w\|_p} + p^2 v_j^{p-2}\left( \frac{[\mathbf{A}(\delta \circ w)]_j}{\|\mathbf{A}w\|_p} \right)^2 \right) \right)^{1/p}.$$

Using $(1+x)^p > 1 + px$, i.e. $(1+px)^{1/p} < 1 + x$, we thus obtain

$$\|\mathbf{A}((1+\delta) \circ w)\|_p \le \|\mathbf{A}w\|_p + \left\langle v^{p-1}, \mathbf{A}(\delta \circ w) \right\rangle + p \left\langle v^{p-1}, \frac{(\mathbf{A}(\delta \circ w))^2}{\mathbf{A}w} \right\rangle.$$

Cauchy-Schwarz yields that $[\mathbf{A}(\delta \circ w)]_j^2 \le [\mathbf{A}(\delta^2 \circ w)]_j [\mathbf{A}w]_j, \forall j \in [d]$. Substituting into the above,

$$\|\mathbf{A}((1+\delta) \circ w)\|_p \le \|\mathbf{A}w\|_p + \left\langle v^{p-1}, \mathbf{A}(\delta \circ w) \right\rangle + p \left\langle v^{p-1}, \mathbf{A}(\delta^2 \circ w) \right\rangle$$

$$= \|\mathbf{A}w\|_p + \sum_{j \in [d]} \left[ \mathbf{A}^\top v^{p-1} \right]_j \delta_j w_j (1 + p\delta_j). \tag{30}$$

Finally, to bound this latter quantity, since $\delta = \eta g$, we observe that for all $j$ either $\delta_j = 0$ or $1 + p\delta_j = 1 + g_j = 2 - [\mathbf{A}^\top v^{p-1}]_j$, in which case

$$\left[ \mathbf{A}^\top v^{p-1} \right]_j (1 + p\delta_j) = \left[ \mathbf{A}^\top v^{p-1} \right]_j \left( 2 - \left[ \mathbf{A}^\top v^{p-1} \right]_j \right) \le 1.$$

Thus, plugging this bound into (30) entrywise,

$$\|\mathbf{A}((1+\delta) \circ w)\|_p - \|\mathbf{A}w\|_p \le \sum_{j \in [d]} \delta_j w_j \left[ \mathbf{A}^\top v^{p-1} \right]_j (1 + p\delta_j) \le \sum_{j \in [d]} \delta_j w_j = \|w_{t+1}\|_1 - \|w_t\|_1.$$

Rearranging yields the desired claim. $\qquad \square$

**Theorem 4.** *Algorithm 2 runs in time $O(\mathrm{nnz}(\mathbf{A}) \cdot \frac{p \log(nd/\epsilon)}{\epsilon})$. Further, its output solves Problem 2.*

*Proof.* The runtime follows from Line 7 (each iteration cost is dominated by multiplication through $\mathbf{A}$), so we prove correctness. Define potential $\Phi_t$ as in Lemma 3, and note that as $w_0 = \frac{\epsilon}{n^2 d}\mathbf{1}$,

$$\Phi_0 \le \|\mathbf{A}w_0\|_p \le \frac{1}{n} \|\mathbf{1}\|_p \le 1.$$

The second inequality followed from our assumption on $\mathbf{A}$ entry sizes (Lemma 14). If Algorithm 2 breaks out of the while loop of Line 4, we have by Lemma 3 that for $x$ returned on Line 11,

$$\|\mathbf{A}w_t\|_p - \|w_t\|_1 \le 1 \implies \|\mathbf{A}x\|_p \le \frac{1 + \|w_t\|_1}{\|w_t\|_1} \le 1 + \epsilon.$$

Thus, primal feasibility is always correct. We now prove correctness of dual feasibility. First, let $V_x(u) = \sum_{i \in [n]} u_i \log(\frac{u_i}{x_i})$ be the Kullback-Leibler divergence from $x$ to $u$, for $x, u \in \Delta^d$. Define the normalized points $x_t = \frac{w_t}{\|w_t\|_1}$ in each iteration. Expanding definitions,

$$V_{x_{t+1}}(u) - V_{x_t}(u) = \sum_{i \in [n]} u_i \log \frac{[x_t]_i}{[x_{t+1}]_i}$$

$$= \sum_{i \in [n]} u_i \left( \log \left( \frac{\|w_{t+1}\|_1}{\|w_t\|_1} \right) + \log \left( \frac{1}{1 + \eta[g_t]_i} \right) \right) \tag{31}$$

$$\le -\eta(1-\eta) \left\langle g_t, u \right\rangle + \log \left( \frac{\|w_{t+1}\|_1}{\|w_t\|_1} \right).$$

The only inequality used the bounds, for $g, \eta \in [0, 1]$,

$$\log\left(\frac{1}{1+\eta g}\right) \leq g \log\left(\frac{1}{1+\eta}\right) \leq -\eta(1-\eta)g.$$

Telescoping (31) over all $T$ iterations, and using $V_{x_0}(u) \leq \log n$ for all $u \in \Delta^n$ since $x_0$ is uniform, we have that whenever Line 4 is not satisfied before the check on Line 7 (i.e. $t \geq T$),

$$\eta(1-\eta) \sum_{0 \leq t < T} \langle g_t, u \rangle \leq \log\left(\frac{\|w_T\|_1}{\|w_0\|_1}\right) + V_{x_0}(u) \leq \log\left(\frac{nd}{\epsilon^2}\right) + \log n \leq 2\log\left(\frac{nd}{\epsilon}\right). \quad (32)$$

The last inequality used $\|w_T\|_1 \leq \epsilon^{-1}$ by assumption, and $\|w_0\|_1 = \frac{\epsilon}{nd}$. Next, since each $g_t \geq \mathbf{1} - \mathbf{A}^\top(v_t)^{p-1}$ entrywise, defining $\bar{z} = \frac{z}{T}$,

$$\sum_{0 \leq t < T} \langle g_t, u \rangle \geq \sum_{0 \leq t < T} \left\langle \mathbf{1} - \mathbf{A}^\top(v_t)^{p-1}, u \right\rangle = T\left\langle \mathbf{1} - \mathbf{A}^\top \bar{z}, u \right\rangle, \text{ for all } u \in \Delta^n. \quad (33)$$

Combining (32) and (33), and rearranging, yields by definition of $T$,

$$\left\langle \mathbf{1} - \mathbf{A}^\top \bar{z}, u \right\rangle \leq \frac{2\log(\frac{nd}{\epsilon})}{T\eta(1-\eta)} \leq \frac{4p\log(\frac{nd}{\epsilon})}{T} \leq \epsilon \implies \mathbf{A}^\top \bar{z} \geq 1 - \epsilon \text{ entrywise.}$$

The last claim follows by setting $u$ to be each coordinate-sparse simplex vector. Finally, since $\frac{\bar{z}}{\|\bar{z}\|_q} = \frac{z}{\|z\|_q}$, to show that $y$ is a correct dual solution to Problem 2 it suffices to show $\|\bar{z}\|_q \leq 1$. This follows as $\bar{z}$ is an average of the $(v_t)^{p-1}$, convexity of $\ell_q$ norms, and that for all $t$,

$$\left\|(v_t)^{p-1}\right\|_q^q = \sum_{j \in [d]} v_t^p = \sum_{j \in [d]} \frac{[\mathbf{A}w_t]_j^p}{\|\mathbf{A}w_t\|_p^p} = 1.$$

$\square$

## C.2 Proofs from Section 4.3

Our analysis of Algorithm 3 will use the following helper fact.

**Lemma 15** (Spectral bounds on $\{\mathbf{A}_i\}_{i \in [n]}$). *Feasibility of Problem 3 is unaffected (up to constants in $\epsilon$) by removing matrices $\mathbf{A}_i$ with an eigenvalue larger than $n\epsilon^{-1}$.*

*Proof.* The proof is identical to Lemma 14; we also require the additional fact that the Schatten norm $\|\cdot\|_p$ is monotone in the Loewner order, forcing the constraint $x_i \leq \frac{\epsilon(1+\epsilon)}{n}$. $\square$

We remark that we can perform this preprocessing procedure via power iteration on each $\mathbf{A}_i$.

**Lemma 4.** *In all iterations $t$ of Algorithm 3, defining $\Phi_t := \left\|\sum_{i \in [n]}[w_t]_i \mathbf{A}_i\right\|_p - \|w_t\|_1$, $\Phi_{t+1} \leq \Phi_t$.*

*Proof.* Drop $t$ and define $\delta = \eta g$. For simplicity, define the matrices

$$\mathbf{M}_0 := \sum_{i \in [n]} w_i \mathbf{A}_i, \ \mathbf{M}_1 := \sum_{i \in [n]} \delta_i w_i \mathbf{A}_i, \ \mathbf{M}_2 := \sum_{i \in [n]} \delta_i^2 w_i \mathbf{A}_i.$$

We recall the Lieb-Thirring inequality $\mathrm{Tr}((\mathbf{ABA})^p) \leq \mathrm{Tr}(\mathbf{A}^{2p}\mathbf{B}^p)$. Applying this, we have

$$\|\mathbf{M}_0 + \mathbf{M}_1\|_p^p = \mathrm{Tr}\left((\mathbf{M}_0 + \mathbf{M}_1)^p\right) \leq \mathrm{Tr}\left(\mathbf{M}_0^p\left(\mathbf{I} + \mathbf{M}_0^{-\frac{1}{2}}\mathbf{M}_1\mathbf{M}_0^{-\frac{1}{2}}\right)^p\right).$$

As $g \leq 1$, we have $\mathbf{M}_0^{-\frac{1}{2}}\mathbf{M}_1\mathbf{M}_0^{-\frac{1}{2}} \preceq p^{-1}\mathbf{I}$. Applying the bounds $(\mathbf{I} + \mathbf{M})^p \preceq \exp(p\mathbf{M}) \preceq \mathbf{I} + p\mathbf{M} + p^2\mathbf{M}^2$ for $\mathbf{M} = \mathbf{M}_0^{-\frac{1}{2}}\mathbf{M}_1\mathbf{M}_0^{-\frac{1}{2}}$, where we use that $\mathbf{I}$ commutes with all $\mathbf{M}$, it follows that

$$\|\mathbf{M}_0 + \mathbf{M}_1\|_p^p \leq \mathrm{Tr}\left(\mathbf{M}_0^p + p\mathbf{M}_0^{p-1}\mathbf{M}_1 + p^2\mathbf{M}_0^{p-1}\mathbf{M}_1\mathbf{M}_0^{-1}\mathbf{M}_1\right).$$

Definitions of $\mathbf{M}_0$, $\mathbf{M}_1$, $\mathbf{M}_2$, and preservation of positiveness under Schur complements imply

$$\begin{pmatrix} \mathbf{M}_0 & \mathbf{M}_1 \\ \mathbf{M}_1 & \mathbf{M}_2 \end{pmatrix} \succeq 0 \implies \mathbf{M}_2 - \mathbf{M}_1 \mathbf{M}_0^{-1} \mathbf{M}_1 \succeq 0.$$

Thus, $\mathbf{M}_1 \mathbf{M}_0^{-1} \mathbf{M}_1 \preceq \mathbf{M}_2$. Applying this and recalling $\mathbf{V} = \frac{\mathbf{M}_0}{\|\mathbf{M}_0\|_p}$,

$$\|\mathbf{M}_0 + \mathbf{M}_1\|_p^p \leq \mathrm{Tr}\left(\mathbf{M}_0^p + p\mathbf{M}_0^{p-1}\mathbf{M}_1 + p^2\mathbf{M}_0^{p-1}\mathbf{M}_2\right)$$

$$= \|\mathbf{M}_0\|_p^p \left(1 + p\left\langle \mathbf{V}^{p-1}, \frac{\mathbf{M}_1}{\|\mathbf{M}_0\|_p} + \frac{p\mathbf{M}_2}{\|\mathbf{M}_0\|_p} \right\rangle\right).$$

By $(1+px)^{1/p} < 1 + x$, taking $p^{th}$ roots we thus have

$$\|\mathbf{M}_0 + \mathbf{M}_1\|_p \leq \|\mathbf{M}_0\|_p + \left\langle \mathbf{V}^{p-1}, \mathbf{M}_1 + p\mathbf{M}_2 \right\rangle.$$

Finally, the conclusion follows as in Lemma 3; by linearity of trace and $g = p\delta$,

$$\left\langle \mathbf{V}^{p-1}, \mathbf{M}_1 + p\mathbf{M}_2 \right\rangle = \sum_{i \in [n]} \left\langle \mathbf{V}^{p-1}, \mathbf{A}_i \right\rangle \delta_i w_i (1 + p\delta_i) \leq \sum_{i \in [n]} \delta_i w_i.$$

Here, we used the inequality for all nonzero $g_i$,

$$\left\langle \mathbf{V}^{p-1}, \mathbf{A}_i \right\rangle (1 + p\delta_i) = \left\langle \mathbf{V}^{p-1}, \mathbf{A}_i \right\rangle \left(2 - \left\langle \mathbf{V}^{p-1}, \mathbf{A}_i \right\rangle\right) \leq 1.$$

$\square$

**Theorem 5.** *Let $p$ be odd. Algorithm 3 runs in $O(\frac{p\log(nd/\epsilon)}{\epsilon})$ iterations, and its output solves Problem 3. Each iteration is implementable in $O(\mathrm{nnz} \cdot \frac{p\log(nd/\epsilon)}{\epsilon^2})$, where $\mathrm{nnz}$ is the number of nonzero entries amongst all $\{\mathbf{A}_i\}_{i \in [n]}$, losing $O(\epsilon)$ in the quality of Problem 3 with probability $1 - \mathrm{poly}((nd/\epsilon)^{-1})$.*

*Proof.* The proof is analogous to that of Theorem 4; we sketch the main differences here. By applying Lemma 15 and monotonicity of Schatten norms in the Loewner order, we again have $\Phi_0 \leq 1$, implying correctness whenever the algorithm terminates on Line 4. Correctness of dual certification again follows from lack of termination and the choice of $T$, as well as setting $u$ to indicate each coordinate. Finally, the returned matrix in Line 8 is correct by convexity of the Schatten-$q$ norm, and the fact that all $\mathbf{V}_t^{p-1}$ have unit Schatten-$q$ norm.

We now discuss issues regarding computing $g_t$ in Line 5 of the algorithm, the bottleneck step; these techniques are standard in the approximate SDP literature, and we defer a more formal discussion to e.g. [JLL+20]. First, note that each coordinate of $g_t$ requires us to compute

$$\frac{1}{\left\|\sum_{i \in [n]} [w_t]_i \mathbf{A}_i\right\|_p^{p-1}} \cdot \left\langle \mathbf{A}_i, \left(\sum_{i \in [n]} [w_t]_i \mathbf{A}_i\right)^{p-1} \right\rangle. \tag{34}$$

We estimate the two quantities in the above expression each to $1 + \epsilon$ multiplicative error with high probability. Union bounding over iterations, and modifying Lemma 4 to use the potential $\left\|\sum_{i \in [n]} [w_t]_i \mathbf{A}_i\right\|_p - (1 + O(\epsilon)) \|w_t\|_1$, the analysis remains valid up to constants in $\epsilon$ with this multiplicative approximation quality. We now discuss our approximation strategies.

For shorthand, denote $\mathbf{M} = \sum_{i \in [n]} [w_t]_i \mathbf{A}_i$. To estimate the denominator of (34), it suffices to multiplicatively approximate $\|\mathbf{M}\|_p^p = \mathrm{Tr}[\mathbf{M}^p]$ within a $1 + \epsilon$ factor, as raising to the $\frac{p-1}{p}$ power can only improve this. To do so, we use the well-known fact (e.g. [DG03]) that letting $\mathbf{Q}$ be a $k \times d$ matrix with independent entries $\sim \mathcal{N}(0, \frac{1}{k})$, for $k = O(\frac{\log(\frac{nd}{\epsilon})}{\epsilon^2})$, with probability $1 - \mathrm{poly}((\frac{nd}{\epsilon})^{-1})$,

$$\mathrm{Tr}[\mathbf{M}^p] \approx \sum_{\ell \in [k]} \mathbf{Q}_{\ell:}^\top \mathbf{M}^p \mathbf{Q}_{\ell:}$$

to a $1 + \epsilon$ factor. To read this from the standard Johnson-Lindestrauss guarantee, it suffices to factorize $\mathbf{M}^p$ and use that each row of the square root's $\ell_2$ norm is preserved with high probability under multiplication by $\mathbf{Q}$, and then apply the cyclic definition of trace. Similarly, for each $i \in [n]$, we can approximate the numerators via

$$\mathrm{Tr}\left(\mathbf{Q}\mathbf{M}^{\frac{p-1}{2}}\mathbf{A}_i\mathbf{M}^{\frac{p-1}{2}}\mathbf{Q}^\top\right).$$

We can simultaneously compute all such quantities by first applying $O(p)$ matrix-vector multiplications through $\mathbf{M}$ to each row of $\mathbf{Q}$, and then computing all quadratic forms. In total, the computational cost per iteration of all approximations is $O(\mathrm{nnz} \cdot \frac{p\log(\frac{nd}{\epsilon})}{\epsilon^2})$ as desired. $\qquad\square$

### C.3 Proof of Proposition 2

In this section, following our prior developments, we prove the following claim.

**Proposition 2.** *Following Theorem 5's notation, let $p$ be odd, $\{\mathbf{A}_i\}_{i\in[n]} \in \mathbb{S}_{\geq 0}^d$, $0 < \epsilon = O(\alpha)$, and*

$$\min_{\substack{x \in \Delta^n \\ \|x\|_\infty \leq \frac{1+\alpha}{n}}} \|\mathcal{A}(x)\|_p = \mathrm{OPT}. \tag{4}$$

*for $\mathcal{A}(x) := \sum_{i\in[n]} x_i \mathbf{A}_i$. Given estimate of $\mathrm{OPT}$ exponentially bounded in $\frac{nd}{\epsilon}$, there is a procedure calling Algorithm 7 $O(\log\frac{nd}{\epsilon})$ times giving $x \in \Delta^n$ with $\|x\|_\infty \leq \frac{(1+\alpha)(1+\epsilon)}{n}$, $\|\mathcal{A}(x)\|_p \leq (1 + \epsilon)\mathrm{OPT}$. Algorithm 7 runs in $O(\frac{\log(nd/\epsilon)\log n}{\epsilon^2})$ iterations, each requiring time $O(\mathrm{nnz} \cdot \frac{p\log(nd/\epsilon)}{\epsilon^2})$.*

#### C.3.1 Reduction to a decision problem

Given access to an oracle for the following approximate decision problem, we can implement an efficient binary search for estimating OPT. Specifically, letting the range of OPT be $(\mu_{\mathrm{lower}}, \mu_{\mathrm{upper}})$, we can subdivide the range into $O(\frac{1}{\epsilon}\log\frac{\mu_{\mathrm{upper}}}{\mu_{\mathrm{lower}}})$ multiplicative intervals of range $1 + \epsilon$, and then compute a binary search using our decision oracle. This incurs a multiplicative $\log(\frac{nd}{\epsilon})$ overhead in the setting of Proposition 2 (see Appendix A, [JLL+20], for a more formal treatment).

**Problem 4.** *Given $\{\mathbf{A}_i\}_{i\in[n]} \in \mathbb{S}_{\geq 0}^d$, either find primal solution $x \in \Delta^n$ with $\|\mathcal{A}(x)\|_p \leq 1 + \epsilon$, $\|x\|_\infty \leq \frac{(1+\epsilon)(1+\alpha)}{n}$, or conclude no $x \in \Delta^n$ satisfies $\|\mathcal{A}(x)\|_p \leq 1 - \epsilon$, $\|x\|_\infty \leq \frac{(1-\epsilon)(1+\alpha)}{n}$.*

The hard constraint $\|x\|_\infty \leq \frac{1+\alpha}{n}$ in the definition (4) can be adjusted by constant factors to admit the $\ell_\infty$ bound in Problem 4, since we assumed $\epsilon = O(\alpha)$ is sufficiently small.

#### C.3.2 Preliminaries

We use the shorthand $\mathbf{S} := \frac{n}{1+\alpha}\mathbf{I}$, and $p' := \frac{\log n}{\epsilon}$, so $\ell_{p'}$ and $\ell_\infty$ are interchangeable up to $1 + O(\epsilon)$ factors. In other words, Problem 4 asks to certify whether there exists $x \in \Delta^n$ with

$$\max\left(\|\mathcal{A}(x)\|_p, \ \|\mathbf{S}x\|_{p'}\right) \leq 1, \tag{35}$$

up to multiplicative $1 + \epsilon$ tolerance on either side. Consider the potential function

$$\Phi(w) := \log\left(\exp\left(\|\mathcal{A}(w)\|_p\right) + \exp\left(\|\mathbf{S}w\|_{p'}\right)\right) - \|w\|_1. \tag{36}$$

It is clear that the first term of $\Phi(w)$ approximates the left hand side of (35) up to a $\log 2$ additive factor, so if any of $\|\mathcal{A}(w)\|_p$, $\|\mathcal{A}(w)\|_{p'}$, or $\|w\|_1$ reaches the scale $3\epsilon^{-1}$ and $\Phi(w)$ is bounded by 1, we can safely terminate. and conclude primal feasibility for Problem 4. Next, we compute

$$\nabla_i \Phi(w) = 1 - \frac{\exp\left(\|\mathcal{A}(w)\|_p\right)\langle\mathbf{A}_i, \mathbf{Y}(w)\rangle + \exp\left(\|\mathbf{S}w\|_{p'}\right)[\mathbf{S}z(w)]_i}{\exp\left(\|\mathcal{A}(w)\|_p\right) + \exp\left(\|\mathbf{S}w\|_{p'}\right)} \quad \text{for all } i \in [n],$$

$$\text{where } \mathbf{Y}(w) := \left(\frac{\mathcal{A}(w)}{\|\mathcal{A}(w)\|_p}\right)^{p-1}, \ z(w) := \left(\frac{\mathbf{S}w}{\|\mathbf{S}w\|_{p'}}\right)^{p'-1} \tag{37}$$

The following helper lemma will be useful in concluding dual infeasibility of Problem 4.

**Lemma 16.** *In the setting of Problem 4, suppose there exists $x^* \in \Delta^n$ with*

$$\|\mathcal{A}(x^*)\|_p \leq 1 - \epsilon, \quad \|\mathbf{S}x^*\|_{p'} \leq 1 - \epsilon.$$

*Then, for any $w$,*

$$\langle \nabla \Phi(w), x^* \rangle \geq \epsilon.$$

*Proof.* From the definitions in (37), it is clear that $\|\mathbf{Y}(w)\|_q = \|z(w)\|_{q'} = 1$, where $q$, $q'$ are the dual norms of $p$, $p'$ respectively. Moreover, by the definition of $x^*$, we have for all $\|\mathbf{Y}\|_q = \|z\|_{q'} = 1$,

$$\langle \mathbf{Y}, \mathcal{A}(x) \rangle \leq 1 - \epsilon, \quad \langle z, \mathbf{S}x \rangle \leq 1 - \epsilon.$$

This follows from the dual definition of the $\ell_p$ norm (see Fact 2). Now, note that for some nonnegative $\alpha(w)$, $\beta(w)$ summing to 1, using the above claim and (37),

$$\langle \nabla \Phi(w), x^* \rangle = 1 - (\alpha(w) \langle \mathbf{Y}(w), \mathcal{A}(x^*) \rangle + \beta(w) \langle z(w), \mathbf{S}x^* \rangle) \geq \epsilon,$$

as desired (here, we used positivity of all relevant quantities). $\qquad\square$

### C.3.3 Potential monotonicity

We prove a monotonicity property regarding the potential $\Phi$ in (36).

**Lemma 17.** *Let $w \in \mathbb{R}_{\geq 0}^n$ satisfy $\|\mathcal{A}(w)\|_p \leq 3\epsilon^{-1}$, $\|\mathbf{S}w\|_{p'} \leq 3\epsilon^{-1}$, let $g = \max(0, \nabla\Phi(w))$ entrywise, and let $w' = (1 + \eta g) \circ w$, where $\eta = (4p')^{-1}$. Then, $\Phi(w') \leq \Phi(w)$.*

*Proof.* Denote for simplicity the threshold $K = 3\epsilon^{-1}$ and the step vector $\delta = \eta g$. First, by prior calculations in Lemma 3 and Lemma 4, it follows that

$$\|\mathcal{A}(w')\|_p \leq \|\mathcal{A}(w)\|_p + \Delta_{\mathcal{A}}, \quad \|\mathbf{S}w'\|_{p'} \leq \|\mathbf{S}w\|_{p'} + \Delta_{\mathbf{S}},$$

where $\Delta_{\mathcal{A}} := \sum_{i \in [n]} \langle \mathbf{A}_i, \mathbf{Y}(w) \rangle \delta_i w_i (1 + p\delta_i)$, $\Delta_{\mathbf{S}} := \sum_{i \in [n]} [\mathbf{S}z(w)]_i \delta_i w_i (1 + p'\delta_i)$.

Next, note that by $\delta \leq \eta$ entrywise and lack of termination (i.e. the threshold $K$),

$$\Delta_{\mathcal{A}} \leq (1 + p\eta)\eta \langle \mathbf{Y}(w), \mathcal{A}(w) \rangle \leq 2\eta \|\mathcal{A}(w)\|_p \leq 1.$$

Therefore, by $\exp(x) \leq 1 + x + x^2$ for $x \leq 1$,

$$\exp\left(\|\mathcal{A}(w')\|_p\right) \leq \exp\left(\|\mathcal{A}(w)\|_p\right)\left(1 + \Delta_{\mathcal{A}} + \Delta_{\mathcal{A}}^2\right). \tag{38}$$

Moreover, by applying Cauchy-Schwarz and the threshold $\|\mathcal{A}(w)\|_p \leq K$ once more,

$$\begin{aligned}
\Delta_{\mathcal{A}}^2 &\leq (1 + p\eta)^2 \left(\sum_{i \in [n]} \langle \mathbf{A}_i, \mathbf{Y}(w) \rangle \delta_i w_i\right)^2 \\
&\leq 2 \left(\sum_{i \in [n]} \langle \mathbf{A}_i, \mathbf{Y}(w) \rangle \delta_i^2 w_i\right) \langle \mathbf{Y}(w), \mathcal{A}(w) \rangle \leq 2K \left(\sum_{i \in [n]} \langle \mathbf{A}_i, \mathbf{Y}(w) \rangle \delta_i^2 w_i\right).
\end{aligned} \tag{39}$$

Combining (38) and (39) (and applying similar reasoning to the term $\Delta_{\mathbf{S}}$), we conclude

$$\exp\left(\|\mathcal{A}(w')\|_p\right) \leq \exp\left(\|\mathcal{A}(w)\|_p\right)\left(1 + \sum_{i \in [n]} \langle \mathbf{A}_i, \mathbf{Y}(w) \rangle \delta_i w_i (1 + (p + 2K)\delta_i)\right),$$

$$\exp\left(\|\mathbf{S}w'\|_{p'}\right) \leq \exp\left(\|\mathbf{S}w\|_{p'}\right)\left(1 + \sum_{i \in [n]} [\mathbf{S}z(w)]_i \delta_i w_i (1 + (p' + 2K)\delta_i)\right).$$

Recall the inequality $\log(1+x) \leq x$ for nonnegative $x$. Expanding the definition of $\Phi$ and $\nabla\Phi$ (cf. (36)), and plugging in the above bounds, we conclude that

$$\Phi(w') - \Phi(w) = \log\left(\frac{\exp\left(\|\mathcal{A}(w')\|_p\right) + \exp\left(\|\mathbf{S}w'\|_{p'}\right)}{\exp\left(\|\mathcal{A}(w)\|_p\right) + \exp\left(\|\mathbf{S}w\|_{p'}\right)}\right) - \langle\delta, w\rangle$$

$$\leq \sum_{i\in[n]} (1 - \nabla_i\Phi(w))\delta_i w_i (1 + (p' + 2K)\delta_i) - \langle\delta, w\rangle$$

$$= \sum_{i\in[n]} \left((1 - \nabla_i\Phi(w))(1 + (p' + 2K)\delta_i) - 1\right)\delta_i w_i.$$

As before, we show that this sum is entrywise nonpositive. For any $i \in [n]$ with $\delta_i \neq 0$, we have

$$(1 - \nabla_i\Phi(w))(1 + (p' + 2K)\delta_i) - 1 = (1 - \nabla_i\Phi(w))(1 + (p' + 2K)\eta\nabla_i\Phi(w)) - 1$$

$$\leq (1 - \nabla_i\Phi(w))(1 + \nabla_i\Phi(w)) - 1 \leq 0,$$

as desired, where we used that $\eta^{-1} \geq p' + 2K$. This yields the conclusion $\Phi(w') \leq \Phi(w)$. $\qquad\square$

### C.3.4 Algorithm and analysis

Finally, we state Algorithm 7 and prove Proposition 2.

---

**Algorithm 7** BoxedSchattenPacking($\{\mathbf{A}_i\}_{i\in[n]}, \epsilon, p, \alpha$)

---

1: **Input:** $\{\mathbf{A}_i\}_{i\in[n]} \in \mathbb{S}_{\geq 0}^d, \epsilon \in [0, \frac{1}{2}], p \geq 2, \alpha \in [0, n-1]$
2: $p' \leftarrow \frac{\log n}{\epsilon}, \mathbf{S} \leftarrow \frac{n}{1+\alpha}\mathbf{I}$
3: $\eta \leftarrow (4p')^{-1}, K \leftarrow 3\epsilon^{-1}, T \leftarrow \frac{6\log(\frac{nd}{\epsilon})}{\eta\epsilon}$
4: $[w_0]_i \leftarrow \frac{\epsilon}{n^2 d}$ for all $i \in [n], t \leftarrow 0$
5: **while** $\|\mathcal{A}(w_t)\|_p, \|\mathbf{S}w_t\|_{p'}, \|w_t\|_1 \leq K$ **do**
6: $\quad g_t \leftarrow \max(0, \nabla\Phi(w_t))$ entrywise, where we use the definition (36)
7: $\quad w_{t+1} \leftarrow w_t \circ (1 + \eta g_t), t \leftarrow t+1$
8: $\quad$ **if** $t \geq T$ **then**
9: $\quad\quad$ **return** Infeasible
10: $\quad$ **end if**
11: **end while**
12: **return** $x = \frac{w_t}{\|w_t\|_1}$

---

*Proof of Proposition 2.* Correctness of the reduction to deciding Problem 4 follows from the discussion in Section C.3.1. Moreover, by the given Algorithm 7, it is clear (following e.g. the preprocessing of Lemma 15) that $\Phi(w_t) \leq 1$ throughout the algorithm, so whenever the algorithm terminates we have primal feasibility. It suffices to prove that whenever the problem admits $x^*$ with

$$\|\mathcal{A}(x^*)\|_p \leq 1 - \epsilon, \quad \|\mathbf{S}x^*\|_{p'} \leq 1 - \epsilon,$$

then the algorithm terminates on Line 5 in $T$ iterations. Analogously to Theorem 4, we have

$$\eta(1-\eta)\sum_{0\leq t<T}\langle g_t, x^*\rangle \leq \log n - \log\|w_0\|_1 + \log\|w_T\|_1 \leq 2\log\left(\frac{nd}{\epsilon}\right) + \log\|w_T\|_1.$$

Next, since $g_t$ is an upwards truncation of $\nabla\Phi(w_t)$, applying Lemma 16 implies that

$$\|w_T\|_1 \geq \exp\left(\frac{\eta\epsilon T}{2} - 2\log\left(\frac{nd}{\epsilon}\right)\right).$$

The conclusion follows by the definition of $T$, as desired. Finally, the iteration complexity follows analogously to the discussion in Theorem 5's proof, where the only expensive cost is estimating coordinates of the $\mathcal{A}$ component of $\nabla\Phi(w_t)$ every iteration. $\qquad\square$

Finally, we remark that by opening up the dual certificates $\mathbf{Y}(w), \mathbf{Z}(w)$ of our mirror descent analysis, we can in fact implement a stronger version of the decision Problem 4 which returns a feasible dual certificate whenever the primal problem is infeasible. We omit this extension for brevity, as it is unnecessary for our applications, but it is analogous to the analysis of Theorem 5.

# D   Deferred proofs from Section 5

## D.1   Proof of Proposition 3

**Proposition 3.** *There is an algorithm* Power *(Algorithm 1, [MM15]), parameterized by $t \in [d]$, tolerance $\tilde{\epsilon} > 0$, $p \geq 1$, and $\mathbf{A} \in \mathbb{S}_{\geq 0}^d$, which outputs orthonormal $\{z_j\}_{j \in [t]}$ with the guarantee*

$$\left.\begin{array}{ll} \left| z_j^\top \mathbf{A}^p z_j - \lambda_j^p(\mathbf{A}) \right| & \leq \tilde{\epsilon}\lambda_j^p(\mathbf{A}) \\ \left| z_j^\top \mathbf{A}^{p-1} z_j - \lambda_j^{p-1}(\mathbf{A}) \right| & \leq \tilde{\epsilon}\lambda_j^{p-1}(\mathbf{A}) \end{array}\right\} \text{ for all } j \in [t]. \tag{6}$$

*Here, $\lambda_j(\mathbf{A})$ is the $j^{th}$ largest eigenvalue of $\mathbf{A}$. The total time required by the method is $O(\mathrm{nnz}(\mathbf{A})\frac{tp \log d}{\varepsilon})$.*

*Proof.* We claim that Algorithm 1 in [MM15] applied to the matrix $\mathbf{A}^p$ with a careful choice of exponent $q$ in their Algorithm 1 yields this guarantee. Specifically, we choose $q_1, q_2$, both of which satisfy the criteria in their main theorem, such that the iterates produced by simultaneous power iteration $\mathbf{M}^p$ with exponent $q_1$ and $\mathbf{M}^{p-1}$ with exponent $q_2$ are identical; it suffices to choose $q$ a multiple of $p(p-1)$. Thus, we can also apply their guarantees to $\mathbf{A}^{p-1}$ and apply a union bound. Notice that their Algorithm 1 also contains some postprocessing to ensure that they obtain singular values in the right space, which is unnecessary for us, as our matrices are Hermitian. $\square$

## D.2   Proof of Lemma 5

**Lemma 5.** *Let $n = \Omega\left(\frac{d + \log \delta^{-1}}{(\epsilon \log \epsilon^{-1})^2}\right)$. With probability $1 - \frac{\delta}{2}$, the uniform distribution over $G$ attains value $(1 + \frac{\tilde{\epsilon}}{2})\|\mathbf{\Sigma}\|_p$ for objective (5), where $\tilde{\epsilon} = C'\epsilon \log \epsilon^{-1}$ for a universal constant $C' > 0$.*

*Proof.* Lemma 8 implies that letting $w^*$ be the uniform distribution over the uncorrupted samples amongst $X_1, \ldots, X_n$, we have with probability at least $1 - \frac{\delta}{2}$, and denoting $\tilde{\epsilon} := 2C_3 \cdot \epsilon \log \epsilon^{-1}$,

$$\left\| \sum_{i \in [n]} w_i^* X_i X_i^\top \right\|_p \leq \left( 1 + \frac{\tilde{\epsilon}}{2} \right) \|\mathbf{\Sigma}\|_p .$$

Therefore, the mixed $\ell_\infty$-$\ell_p$ packing semidefinite program

$$\exists w^* \in \Delta^n \text{ with } \|w^*\|_\infty \leq \frac{1}{(1-\epsilon)n}, \left\| \sum_{i \in [n]} w_i^* X_i X_i^\top \right\|_p \leq \left( 1 + \frac{\tilde{\epsilon}}{2} \right) \|\mathbf{\Sigma}\|_p$$

is feasible. This completes the proof. $\square$

## D.3   Proof of Lemma 6

**Lemma 6.** *Let $n = \Omega\left(\frac{d + \log \delta^{-1}}{(\epsilon \log \epsilon^{-1})^2}\right)$. With probability at least $1 - \frac{\delta}{2}$, $(1 + \tilde{\epsilon})\mathbf{\Sigma} \succeq \mathbf{M}_G \succeq (1 - \tilde{\epsilon})\mathbf{\Sigma}$.*

*Proof.* We follow the notation of (7). First, by the guarantees of Corollary 1,

$$w_G = 1 - w_B \geq 1 - \frac{\epsilon n}{(1 - 2\epsilon)n} = \frac{1 - 3\epsilon}{1 - 2\epsilon} \geq 1 - 2\epsilon.$$

Therefore, again applying Corollary 1, for all $i \in G$,

$$\frac{w_i}{w_G} \leq \frac{1}{(1 - 2\epsilon)n} \cdot \frac{1 - 2\epsilon}{1 - 3\epsilon} = \frac{1}{(1 - 3\epsilon)n}.$$

We conclude that the set of weights $\{\frac{w_i}{w_G}\}_{i \in G}$ belong to $\mathfrak{S}_{3\epsilon}^{(1-\epsilon)n}$. By applying Corollary 3 to these weights and adjusting the definition of $C_3$ by a constant, we conclude with probability at least $1 - \frac{\delta}{2}$

$$\left( 1 + C_3 \cdot \epsilon \log \epsilon^{-1} \right) \mathbf{\Sigma} \succeq \sum_{i \in G} \frac{w_i}{w_G} X_i X_i^\top \succeq \left( 1 - C_3 \cdot \epsilon \log \epsilon^{-1} \right) \mathbf{\Sigma}.$$

The conclusion follows by multiplying through by $w_G$, and using the definition $\tilde{\epsilon} = 2C_3 \cdot \epsilon \log \epsilon^{-1}$. $\square$

### D.4 Proof of Proposition 4

**Proposition 4.** *Let* $\mathbf{M} = \mathbf{M}_G + \mathbf{M}_B$ *be so that* $\|\mathbf{M}\|_p \leq (1 + \tilde{\epsilon}) \|\mathbf{\Sigma}\|_p$, $\mathbf{M}_G \succeq 0$ *and* $\mathbf{M}_B \succeq 0$, *and so that* $(1 + \tilde{\epsilon})\mathbf{\Sigma} \succeq \mathbf{M}_G \succeq (1 - \tilde{\epsilon})\mathbf{\Sigma}$. *Following notation of Algorithm 4, let*

$$\mathbf{M} = \sum_{j \in [d]} \lambda_j v_j v_j^\top, \quad \mathbf{\Sigma} = \sum_{j \in [d]} \sigma_j u_j u_j^\top \tag{8}$$

*be sorted eigendecompositions of* $\mathbf{M}$ *and* $\mathbf{\Sigma}$, *so* $\lambda_1 \geq \ldots \geq \lambda_d$, *and* $\sigma_1 \geq \ldots \geq \sigma_d$. *Let* $\gamma$ *be as in Theorem 2, and assume* $\sigma_{t+1} < (1 - \gamma)\sigma_1$. *Then,*

$$\max_{j \in [t]} v_j^\top \mathbf{\Sigma} v_j \geq (1 - \gamma) \|\mathbf{\Sigma}\|_\infty.$$

*Proof.* For concreteness, we will define the parameters

$$p = \frac{2}{7} \sqrt{\frac{\log(3d)}{\tilde{\epsilon}}}, \quad \gamma = 14\sqrt{\tilde{\epsilon} \log(3d)} = 49 p \tilde{\epsilon}.$$

For these choices of $p$, $\gamma$, we will use the following (loose) approximations for sufficiently small $\tilde{\epsilon}$:

$$\left(1 - \frac{\gamma}{4}\right)^p = \left(1 - \frac{\gamma}{4}\right)^{\frac{4}{\gamma} \log(3d)} \leq \frac{1}{3d}, \quad (1 + \tilde{\epsilon})^p - (1 - \tilde{\epsilon})^p \leq \exp(p\tilde{\epsilon}) - (1 - p\tilde{\epsilon}) \leq 3p\tilde{\epsilon}. \tag{40}$$

Suppose for contradiction that all $v_j^\top \mathbf{\Sigma} v_j < (1 - \gamma)\sigma_1$ for $j \in [t]$. By applying the guarantee of Corollary 1 and Fact 2, it follows that

$$\langle \mathbf{M}, \mathbf{M}^{p-1} \rangle = \|\mathbf{M}\|_p^p \leq (1 + \tilde{\epsilon})^p \|\mathbf{\Sigma}\|_p^p. \tag{41}$$

Let $s \in [d]$ be the largest index such that $\sigma_s > \left(1 - \frac{\gamma}{4}\right)\sigma_1$, and note that $s \leq t$. We define

$$\mathbf{N} := \sum_{j \in [s]} \lambda_j^{p-1} v_j v_j^\top \preceq \mathbf{M}^{p-1}.$$

That is, $\mathbf{N}$ is the restriction of $\mathbf{M}^{p-1}$ to its top $s$ eigendirections. Then,

$$\begin{aligned}
\langle \mathbf{M}, \mathbf{M}^{p-1} \rangle &= \langle \mathbf{M}_B, \mathbf{M}^{p-1} \rangle + \langle \mathbf{M}_G, \mathbf{M}^{p-1} \rangle \\
&\geq \langle \mathbf{M}_B, \mathbf{M}^{p-1} \rangle + \langle (1 - \tilde{\epsilon})\mathbf{\Sigma}, \mathbf{M}^{p-1} \rangle \geq \langle \mathbf{M}_B, \mathbf{N} \rangle + (1 - \tilde{\epsilon})^p \|\mathbf{\Sigma}\|_p^p.
\end{aligned} \tag{42}$$

In the second line, we used Lemma 6 twice, as well as the trace inequality Lemma 18 with $\mathbf{A} = \mathbf{M}$ and $\mathbf{B} = (1 - \tilde{\epsilon})\mathbf{\Sigma}$. Combining (41) with (42), and expanding the definition of $\mathbf{M}_B$, yields

$$\begin{aligned}
((1 + \tilde{\epsilon})^p - (1 - \tilde{\epsilon})^p) \|\mathbf{\Sigma}\|_p^p &\geq \langle \mathbf{M}_B, \mathbf{N} \rangle = \left\langle \mathbf{M}_B, \sum_{j \in [s]} \lambda_j^{p-1} v_j v_j^\top \right\rangle \\
&= \left\langle \mathbf{M}, \sum_{j \in [s]} \lambda_j^{p-1} v_j v_j^\top \right\rangle - \left\langle \mathbf{M}_G, \sum_{j \in [s]} \lambda_j^{p-1} v_j v_j^\top \right\rangle \\
&\geq \left\langle \mathbf{M}, \sum_{j \in [s]} \lambda_j^{p-1} v_j v_j^\top \right\rangle - (1 + \tilde{\epsilon}) \left\langle \mathbf{\Sigma}, \sum_{j \in [s]} \lambda_j^{p-1} v_j v_j^\top \right\rangle \\
&= \sum_{j \in [s]} \left( \lambda_j^p - (1 + \tilde{\epsilon})\lambda_j^{p-1} v_j^\top \mathbf{\Sigma} v_j \right) \geq \sum_{j \in [s]} \left( \lambda_j^p - \lambda_j^{p-1}(1 + \tilde{\epsilon})(1 - \gamma)\sigma_1 \right).
\end{aligned} \tag{43}$$

The third line followed from from the spectral bound $\mathbf{M}_G \preceq (1 + \tilde{\epsilon})\mathbf{\Sigma}$ of Lemma 6, and the fourth followed from the fact that $\{\lambda_j\}_{j \in [d]}$, $\{v_j\}_{j \in [d]}$ eigendecompose $\mathbf{M}$, as well as the assumption $v_j^\top \mathbf{\Sigma} v_j \leq (1 - \gamma)\sigma_1$ for all $j \in [t]$. Letting $S := \sum_{j \in [s]} \sigma_j^p$, and using both approximations in (40),

$$\|\mathbf{\Sigma}\|_p^p \leq \sum_{j \in [s]} \sigma_j^p + \left(1 - \frac{\gamma}{4}\right)^p (d - s)\sigma_1^p \leq \frac{4}{3} S \implies ((1 + \tilde{\epsilon})^p - (1 - \tilde{\epsilon})^p) \|\mathbf{\Sigma}\|_p^p \leq 4p\tilde{\epsilon} S. \tag{44}$$

Next, we bound the last term of (43). By using $(1 + \tilde{\epsilon})(1 - \gamma) \leq 1 - \frac{\gamma}{2}$,

$$\sum_{j \in [s]} \left( \lambda_j^p - \lambda_j^{p-1}(1 + \tilde{\epsilon})(1 - \gamma)\sigma_1 \right) \geq \sum_{j \in [s]} \lambda_j^{p-1} \left( \lambda_j - \left(1 - \frac{\gamma}{2}\right)\sigma_1 \right)$$

$$\geq \frac{\gamma}{6} \sum_{j \in [s]} \lambda_j^{p-1}\sigma_1 \geq \frac{\gamma}{6}(1 - \tilde{\epsilon})^{p-1} \sum_{j \in [s]} \sigma_j^p \geq \frac{\gamma}{12}S. \tag{45}$$

The second line used $\lambda_j - (1 - \frac{\gamma}{2})\sigma_1 \geq (1 - \tilde{\epsilon})\sigma_j - (1 - \frac{\gamma}{2})\sigma_1 \geq \frac{\gamma}{6}\sigma_1$ by definition of $s$, Lemma 19 (twice), and $(1 - \tilde{\epsilon})^{p-1} \geq \frac{1}{2}$. Combining (45) and (44) and plugging into (43),

$$4p\tilde{\epsilon}S \geq \frac{\gamma}{12}S \implies 48p\tilde{\epsilon} \geq \gamma.$$

By the choice of $\gamma$ and $p$ (i.e. $\gamma = 49p\tilde{\epsilon}$), we attain a contradiction. $\qquad\square$

In the proof of Proposition 4, we used the following facts.

**Lemma 18.** *Let* $\mathbf{A} \succeq \mathbf{B} \succeq 0$ *be symmetric matrices and* $p$ *a positive integer. Then we have*

$$\text{Tr}\left(\mathbf{A}^{p-1}\mathbf{B}\right) \geq \text{Tr}\left(\mathbf{B}^p\right).$$

*Proof.* For any $1 \leq k \leq p - 1$,

$$\text{Tr}\left(\mathbf{A}^k\mathbf{B}^{p-k}\right) \geq \text{Tr}\left(\mathbf{A}^{k-1}\mathbf{B}^{\frac{p-k}{2}}\mathbf{A}\mathbf{B}^{\frac{p-k}{2}}\right) \geq \text{Tr}\left(\mathbf{A}^{k-1}\mathbf{B}^{\frac{p-k}{2}}\mathbf{B}\mathbf{B}^{\frac{p-k}{2}}\right) = \text{Tr}\left(\mathbf{A}^{k-1}\mathbf{B}^{p-k+1}\right).$$

The first step used the Extended Lieb-Thirring trace inequality $\text{Tr}(\mathbf{M}\mathbf{N}^2) \geq \text{Tr}(\mathbf{M}^\alpha \mathbf{N}\mathbf{M}^{1-\alpha}\mathbf{N})$ for $\alpha \in [0, 1]$, $\mathbf{M}, \mathbf{N} \in \mathbb{S}_{\geq 0}^d$ (see e.g. Lemma 2.1, [ALO16]), and the second $\mathbf{A} \succeq \mathbf{B}$. Finally, induction on $k$ yields the claim. $\qquad\square$

**Lemma 19.** *For all* $j \in [d]$, $\lambda_j \geq (1 - \tilde{\epsilon})\sigma_j$.

*Proof.* By the Courant-Fischer minimax characterization of eigenvalues,

$$\lambda_j \geq \min_{k \in [j]} u_k^\top \mathbf{M} u_k.$$

However, we also have $\mathbf{M} \succeq \mathbf{M}_G \succeq (1 - \tilde{\epsilon})\mathbf{\Sigma}$ (Lemma 6), yielding the conclusion. $\qquad\square$

### D.5 Proof of Theorem 2

The guarantees of Proposition 4 were geared towards exact eigenvectors of the matrix $\mathbf{M}$. We now modify the analysis to tolerate inexactness in the eigenvector computation, in line with the processing of Line 5 of our Algorithm 4. This yields our final claim in Theorem 2.

**Corollary 5.** *In the setting of Proposition 4, and letting* $\{z_j\}_{j \in [t]}$ *satisfy* (6), *set for all* $j \in [t]$

$$y_j := \frac{\mathbf{M}^{\frac{p-1}{2}} z_j}{\left\| \mathbf{M}^{\frac{p-1}{2}} z_j \right\|_2}.$$

*Then with probability at least* $1 - \delta$,

$$\max_{j \in [t]} y_j^\top \mathbf{\Sigma} y_j \geq (1 - \gamma) \|\mathbf{\Sigma}\|_\infty.$$

*Proof.* Assume all $y_j$ have $y_j^\top \mathbf{\Sigma} y_j \leq (1 - \gamma)\sigma_1$ for contradiction. We outline modifications to the proof of Proposition 4. Specifically, we redefine the matrix $\mathbf{N}$ by

$$\mathbf{N} := \mathbf{M}^{\frac{p-1}{2}} \left( \sum_{j \in [s]} z_j z_j^\top \right) \mathbf{M}^{\frac{p-1}{2}}.$$

Because $\sum_{j \in [s]} z_j z_j^\top$ is a projection matrix, it is clear $\mathbf{N} \preceq \mathbf{M}^{p-1}$. Therefore, by combining the derivations (41) and (42), it remains true that

$$((1 + \tilde{\epsilon})^p - (1 - \tilde{\epsilon})^p) \|\mathbf{\Sigma}\|_p^p \geq \langle \mathbf{M}_B, \mathbf{N} \rangle = \langle \mathbf{M}, \mathbf{N} \rangle - \langle \mathbf{M}_G, \mathbf{N} \rangle.$$

We now bound these two terms in an analogous way from Proposition 4, with negligible loss; combining these bounds will again yield a contradiction. First, we have the lower bound

$$\left\langle \mathbf{M}, \sum_{j \in [s]} \mathbf{M}^{\frac{p-1}{2}} z_j z_j^\top \mathbf{M}^{\frac{p-1}{2}} \right\rangle = \sum_{j \in [s]} z_j^\top \mathbf{M}^p z_j \geq (1 - \tilde{\epsilon}) \sum_{j \in [s]} \lambda_j^p.$$

Here, the last inequality applied the assumption (6) with respect to $\mathbf{M}^p$. Next, we upper bound

$$
\begin{aligned}
\left\langle \mathbf{M}_G, \sum_{j \in [s]} \mathbf{M}^{\frac{p-1}{2}} z_j z_j^\top \mathbf{M}^{\frac{p-1}{2}} \right\rangle &\leq (1 + \tilde{\epsilon}) \left\langle \mathbf{\Sigma}, \sum_{j \in [s]} \mathbf{M}^{\frac{p-1}{2}} z_j z_j^\top \mathbf{M}^{\frac{p-1}{2}} \right\rangle \\
&= (1 + \tilde{\epsilon}) \sum_{j \in [s]} \left\| \mathbf{M}^{\frac{p-1}{2}} z_j \right\|_2^2 y_j^\top \mathbf{\Sigma} y_j \\
&\leq (1 + \tilde{\epsilon})(1 - \gamma) \sigma_1 \sum_{j \in [s]} z_j^\top \mathbf{M}^{p-1} z_j \\
&\leq (1 - \gamma)(1 + \tilde{\epsilon})^2 \sigma_1 \sum_{j \in [s]} \lambda_j^{p-1},
\end{aligned}
$$

The first line used $\mathbf{M}_G \preceq (1 + \tilde{\epsilon})\mathbf{\Sigma}$, the second used the definition of $y_j$, the third used our assumption $y_j^\top \mathbf{\Sigma} y_j \leq (1 - \gamma)\sigma_1$, and the last used (6) with respect to $\mathbf{M}^{p-1}$. Finally, the remaining derivation (45) is tolerant to additional factors of $1 + \tilde{\epsilon}$, yielding the same conclusion up to constants. $\qquad\square$

Finally, we prove Theorem 2 by combining the tools developed thus far.

*Proof of Theorem 2.* Correctness of the algorithm is immediate from Corollary 5 and the guarantees of 1DRobustVariance. Concretely, Corollary 5 guarantees that one of the vectors we produce will be a $(1 - \gamma)$-approximate top eigenvector (say some index $j \in [t]$), and 1DRobustVariance will only lose a negligible fraction $O(\epsilon \log \epsilon^{-1})$ of this quality (see Lemma 1); the best returned eigenvector as measured by 1DRobustVariance can only improve the guarantee. Finally, the failure probability follows by combining the guarantees of Lemmas 1, 5, and 6.

We now discuss runtime. The complexity of lines 2, 4, and 5, as guaranteed by Proposition 2, Proposition 3, and Lemma 1 are respectively (recalling $p = \widetilde{O}(\epsilon^{-0.5})$)

$$\widetilde{O}\left(\frac{nd}{\epsilon^{4.5}}\right), \ \widetilde{O}\left(\frac{ndt}{\epsilon^{1.5}}\right), \ \widetilde{O}(ndt).$$

Throughout we use that we can compute matrix-vector products in an arbitrary linear combination of the $X_i X_i^\top$ in time $O(nd)$; it is easy to check that in all runtime guarantees, nnz can be replaced by this computational cost. Combining these bounds yields the final conclusion. $\qquad\square$