[Reviews · NeurIPS 2020]

Review 1

Summary and Contributions: This paper studied Robust sub-Gaussian principal component analysis problem and Width-independent Schatten packing SDPs problem. The paper gave novel efficient algorithms for both problems.

Strengths: This paper shows several crucial primitives. For robust sub-gaussian problem, authors proposed an algorithm which uses samples (with corruptions) to estimate the variance for any given unit direction.The algorithm is very clean and simple. The authors show that one just needs to drop the most effective O(eps) samples and the remaining samples would give a good approximation. For L_p packing problem, author gave a crucial property of the potential function in the mirror descent procedure.

Weaknesses: My only concern is that the studied two problems are somewhat independent. It is not clear to me what the connection is.

Correctness: I did not see any obvious issues in the proofs.

Clarity: The paper is well written in general.

Relation to Prior Work: This paper has a detailed description of previous literature.

Reproducibility: Yes

Additional Feedback: - line 73, power method for PCA d^2 -> d^3 - line 191, Oour -> our - Section 4.1, good to mention it is also multiplicative weights update - It would be good that authors explain the intrinsic connections between these two problems explicitly. Post rebuttal: Authors addressed my concern. I tend to accept the paper.


Review 2

Summary and Contributions: The paper proposes to estimate the Eigenvector of covariance with sub-Gaussian concentration. Two new algorithms are presented to handle this problem in polynomial time and in nearly-linear time. Besides, a key technical tool, Width-independent Schatten packing SDPs, is developed for the nearly-linear time method.

Strengths: The strengths of this work including proposing two new methods to estimate the top eigenvector of the covariance with sub-Gaussian concentration, a theoritical analysis on the time-complexity of the new algorithms. and a novel width-independent iterative methods.

Weaknesses: /

Correctness: The claims and method seems to be correct.

Clarity: Yes

Relation to Prior Work: Yes

Reproducibility: Yes

Additional Feedback:


Review 3

Summary and Contributions: This paper studies the problem of Robust PCA of a subgaussian distribution. Specifically, one is given samples X_1,X_2,...,X_n from a subgaussian distribution, such that an eps-fraction of the samples have been arbitrarily corruption (modified by an adversary), the goal is to approximately recover the top singular vector of the covariance matrix Sigma. Here, approximately recover means to find a vector u such that u^T Sigma u > (1 - gamma )|Sigma|_op, where |Sigma|_op is the operator norm of Sigma. The main result of this paper are runtime and sample complexity efficient algorithms for this task. Specifically, they show 1) An algorithm that achieves error gamma = O(eps log(1/eps)) in polynomial time, specifically in tilde{O}(n d^2/eps) time, using n > Omega(d /*(eps log(1/eps))^2) samples. 2) Under the assumption that the covariance matrix has an spectral gap between its first and second largest singular values, an algorithm that achieves slightly worse error gamma = O(\sqrt{eps log(1/eps) \log d }) in "nearly-linear" time using the same number of samples. Specifically in tilde{O}(n d/eps^{4.5}) time, using n > Omega(d /*(eps log(1/eps))^2) samples. Prior to this work, the problem of robust covariance estimation and robust PCA were very well studied, however most results (such as those using SoS-based techniques) required more structure on the moments of the distributions than a purely sub-Gaussian guarantee. This is the first such result on robust PCA in the eps-contamination model which gives polynomial time guarantees for general subgaussian distributions. Most prior works obtained guarantees for PCA by first robustly estimating the covariance matrix, which the authors point out is a potentially lossy step. Instead, the authors circumvent the step of covariance estimation, and instead directly estimate the top singular vector. This allows the authors to obtain improved sample complexity (linear in d, as opposed to the d^2 required by known algorithms for learning covariance matrices to spectral norm error). To obtain the result in 2), the authors need to solve the intermediate problem of width-independent Schatten packing. Here, the Schatten packing SDP is to solve min_w |sum_i w_i A_i|_p subject to w \in Delta^n Where Delta^n is the subset of probability distributions (w_1,...,w_n) over n points, A_i's are fixed, and | |_p is the Schatten p-norm. This problem can be solved with additive error depending linearly on the "width" of the problem by an SDP solver. Here, the width rho is the largest spectral norm of a constraint. However, obtaining bounds independent of rho (or logarithmic in rho) were only known for the special case of p=infty (which is the spectral norm). The authors solve this intermediate problem, by providing an algorithm which gives a (1+eps) multiplcative approximation of the objective value with O(p log(nd/eps)/eps) iterations. Thus, the number of iterations required by the solver to obtain relative error is independent of the width of the program. Moreover, for odd p, each iteration can be carried out in linear time. The algorithm proceeds by assigning weights w to samples, initialized to be uniform. One can then compute the top eigenvector u of the empirical covariance with respect to the samples X_1,...,X_n and the weights w_1,...,w_n. However, in the eps-contaminated model, it is easy to see that in general u may not yield a good approximation for u^T Sigma u. Firstly, using a standard procedure, the one can robustly estimate the size of u^T Sigma u from the given samples in the fixed direction u. If this contribution is large enough, then u is a good target direction. Otherwise, they proceed by reweighting the distribution w so as to down-weight potentially corrupted samples. They then compute the new top eigenvector with respect to the new weights, and iteratively continue from there. For their nearly linear time algorithm, to implement the second step of down-weighing the samples efficiently, the authors utilize their width-independent Schatten packing algorithm. %%%%% Post-Rebuttal %%%% I appreciate the author's replies, and continue to think that this is a very strong paper which I believe is well above the threshold to be accepted to NeurIPS. With regards to the response, I would clarify that I am not suggesting that a experimental evaluation should be necessary to add to this paper, just that a discussion on its implementation (that it can be parallelized, ect.) should be highlighted slightly more given the venue.

Strengths: This is a strong result, which should be particularly appealing to the NeurIPS community due to the simplicity of the assumption: only sub-Gaussianity is required (as opposed to more complicated algebraic assumptions on the moments of the distribution, which may not be as easy to justify in practice). The paper employs a mixture of known as well as novel techniques; the approach of re-weighing samples iteratively via an SDP is particularly interesting, and may be inspiring for future algorithms in robust statistics. Improved solvers for SDP packing under different norms (e.g. Schatten and Ky-fan) have emerged as interesting and important problems for a variety of statistical problems. Thus, the Schatten packing solver, while used mainly as an intermediate step for the purposes of the near-linear time algorithm, is indeed interesting as an independent result, and likely useful for other applications.

Weaknesses: While theoretically interesting and technically novel, it's unclear whether the new SDP solver has much practical relevance. Thus, while a key selling point of this paper to the NeurIPS community may be its generality (only requiring sub-gaussian moments), the practical applicability of the algorithm itself is somewhat lacking.

Correctness: Yes (checked main components of proofs).

Clarity: Yes, the paper is well written, and the main contributions and distinctions from prior work are easy to follow.

Relation to Prior Work: Yes (as described in summary).

Reproducibility: Yes

Additional Feedback: The high level overview of the main steps in the algorithm in Section 3 feels fairly abbreviated (even for inclusion in an 8 page conference version). This section should give a very clear picture of which steps were standard in the robust statistics literature, and which steps required substantially new work on the part of the authors, and lastly where the key component of Schatten packing comes in (this fact can be interfered, but slightly more discussion may be useful). For example, the idea of downweighing was not new in this paper (as the authors point out), however their usage of a Schatten packing SDP to improve the runtime of this step appears to be. This particular relationship with prior work should be clarified. Moreover, if this is the main new technical insight of this work, then since the improved SDP solver is not needed for the first (polynomial time) result, the technical novelty of this result should also be clarified.


Review 4

Summary and Contributions: The paper considers the problem of approximating the top eigenvector of covariance matrices when data are sub-Gaussian and \epsilon-corrupted. It proposes two approximation algorithms, one running in polynomial time and the other in nearly linear time, and whose analysis do not require additional structures of moments. The second algorithm is made possible by a novel width-independent SDP solver. Analysis of approximation factors and runtime complexity is provided for both proposed algorithms.

Strengths: The exposition of the paper is very clear, and the technical details are sound. The proposed robust algorithms for PCA, especially that they are provably correct without artificial structures on moments, is a significant contribution to the ML community. The proposed width-independent SDP solver is also an important development for robust learning and other areas of ML.

Weaknesses: - The transition between Sections 3 and 4 is quite abrupt. It would be great if the authors could discuss how Schatten packing can be applied to the robust PCA problem in a high level at the beginning of Section 4; - I would like to see some discussions on the space complexity of the proposed SDP solver, and a comparison against other methods; - Some numerical experiments would be helpful to demonstrate the effectiveness and practicality of the proposed algorithms.

Correctness: Yes

Clarity: Yes

Relation to Prior Work: Yes

Reproducibility: Yes

Additional Feedback: - L5-6, “Our second” misses “algorithm” or “method” - L137, “distribution a” -> “a distribution” - L141, “B, is” -> “B is” - L191, “Oour” -> “Our” - L199, \circ does not seem to have been defined ************** Re rebuttal ************** I appreciate the rebuttal and acknowledge that my concerns have been addressed.

[Author Response · NeurIPS 2020]

We thank all the reviewers for their valuable feedback. Regarding implementation and practicality, this work primarily aimed to address outstanding theoretical questions in the robust statistics and continuous optimization literatures on several fronts. The question of practicality of more standard filtering-based methods (which our first PCA algorithm is an example of), in terms of dimension-dependent bounds on the number of iterations, led us to the development of our second robust PCA algorithm (Algorithm 3). As each of its key subroutines is based entirely on matrix-vector multiplications, and it runs in nearly-linear time for covariance matrices with a mild spectral gap, our hope is that it can indeed find use in practice. We agree it is interesting and important future work to evaluate our methods empirically, but believe it is out of the scope of the current paper as its focus was establishing various theoretical primitives.

**Reviewer 1.** Regarding the relationship between the two problems studied: we give two algorithms for the robust PCA problem; one is developed in Section 3 (Theorem 1), and the other in Section 5 (Theorem 2). The algorithm in Section 5 uses our Schatten packing procedure in Section 4 as a crucial subroutine (see Line 2 of Algorithm 3, and Proposition 2); its development appears to us to be necessary compared to standard operator norm packing procedures, as one needs to reason about the amount of adversarial perturbation along multiple directions. We will make this more clear in the revision. The power method takes $\tilde{O}(d^2)$ with high probability as it can be implemented with logarithmically many matrix-vector multiplies. We agree with your remaining comments and thank you for your suggestions.

**Reviewer 2.** Thank you for your reading. We believe our work addresses an important outstanding question in the robust statistics literature by proposing an efficient algorithm, develops a useful optimization primitive, and makes progress towards the broader goal of covariance estimation under weaker assumptions than is known. We hope you are inclined to raise your opinion of our paper.

**Reviewer 3.** Since our algorithm is conceptually simple and its implementation only requires (parallelizable) matrix-vector multiplies, we are optimistic it (or variants of it) can find use in practice, but acknowledge your concerns. We agree that the algorithm and analysis structure of Section 3 more heavily uses known techniques in the literature (i.e. filtering via downweighting), but will clarify this more directly in the text. On the other hand, our use of a Schatten packing subroutine in Algorithm 3 (as well as using its guarantees in the analysis) is novel, and we hope it finds additional use in robust statistics tasks where operator norm conditions do not suffice, and also as an independently interesting optimization primitive. We appreciate that you suggest this as well; thank you for your detailed comments.

**Reviewer 4.** We will add additional expositional text discussing the utility of a Schatten packing subroutine for the robust PCA task; at a high level, an operator norm packing algorithm cannot detect an adversary which raises a different eigendirection to have a quadratic form equal to the top, which can fool a PCA algorithm. Regarding space complexity: none of our algorithms require more than storing a constant number of additional vectors, i.e. $O(n)$ additive memory overhead, at any point; we will make a note of this. This space complexity overhead is standard for approximate SDP algorithms. The $\circ$ notation means entrywise product; we will clarify this. We agree with your other comments and will revise appropriately; thank you for your careful reading.

[Meta-Review · NeurIPS 2020]

The paper studies principal component analysis for sub-Gaussian distributions. Two new methods for this problem are proposed, one of which uses width-independent Schatten packing SDPs. Reviewers agree that this is an interesting, non-trivial and solid theoretical work and should be accepted for NeurIPS. The rebuttal addressed the reviewers concerns adequately. The recommendation is to accept this paper for presentation at NeurIPS. We urge the authors to make the connection of the Schatten packing to the main approach more clearer in a final version of the paper.